



# WOMBAT v1.0: A fully Bayesian global flux-inversion framework

Andrew Zammit-Mangion[1,*], Michael Bertolacci[1,*], Jenny Fisher[2], Ann Stavert[3], Matthew L. Rigby[4], Yi Cao[1], and Noel Cressie[1]

[1]School of Mathematics and Applied Statistics, University of Wollongong, Wollongong, Australia
[2]School of Earth and Life Sciences, University of Wollongong, Wollongong, Australia
[3]Climate Science Centre, CSIRO Oceans and Atmosphere, Aspendale, Australia
[4]School of Chemistry, University of Bristol, Bristol, UK
[*]These authors contributed equally to this work.

**Correspondence:** Andrew Zammit-Mangion (azm@uow.edu.au)

**Abstract.** WOMBAT (the WOllongong Methodology for Bayesian Assimilation of Trace-gases) is a fully Bayesian hierarchical statistical framework for flux inversion of trace gases from flask, in situ, and remotely sensed data. WOMBAT extends the conventional Bayesian-synthesis framework through the consideration of a correlated error term, the capacity for online bias correction, and the provision of uncertainty quantification on all unknowns that appear in the Bayesian statistical model. We

show, in an observing system simulation experiment (OSSE), that these extensions are crucial when the data are indeed biased and have errors that are spatio-temporally correlated. Using the GEOS-Chem atmospheric transport model, we show that WOMBAT is able to obtain posterior means and variances on non-fossil-fuel $CO_2$ fluxes from Orbiting Carbon Observatory-2 (OCO-2) data that are comparable to those from the Model Intercomparison Project (MIP) reported in Crowell et al. (2019, *Atmos. Chem. Phys.*, vol. 19). We also find that WOMBAT's predictions of out-of-sample retrievals obtained from the Total

Column Carbon Observing Network are, for the most part, more accurate than those made by the MIP participants.

## 1 Introduction

Atmospheric carbon dioxide ($CO_2$) is a leading driver of global warming (Peters et al., 2013). If left unchecked, the rise in global temperatures will have a substantial negative impact on society and the environment (Edenhofer et al., 2014). As part of the worldwide effort to limit these impacts, the 2015 Paris Agreement under the United Nations Framework Convention on

Climate Change, COP 21, called for a global stocktake of the sources and sinks of $CO_2$ and other greenhouse gases, with the first evaluation planned for 2023 (UNFCCC, 2015). The rate at which $CO_2$ is emitted (from sources) or absorbed (at sinks) per unit space and time is known as the $CO_2$ flux, which itself varies spatio-temporally in a substantial manner. Despite the fact that it is human emissions that are driving the rise in atmospheric $CO_2$ concentrations, the most uncertain aspects of quantifying $CO_2$ fluxes at Earth's surface centre around natural processes. For example, while we know that the land and oceans absorb

more than half of the $CO_2$ emitted by human activities (Dlugokencky and Tans, 2020), the geographical and temporal patterns of these sinks remain unclear (Crowell et al., 2019).

Monitoring the progression of $CO_2$ in our atmosphere is thus of utmost importance, and billions of dollars have been spent over the last few decades on research, development, and deployment of instruments for measuring $CO_2$ mole fraction (defined



here as the proportion of $CO_2$ molecules in a given parcel of dry air) throughout the globe (Burrows et al., 1995; Kuze et al., 2009; Wunch et al., 2011a; Masarie et al., 2014; Eldering et al., 2017, 2019). However, $CO_2$ mole fraction is only indirectly related to the key quantity of interest, namely the geographic distribution of the $CO_2$ fluxes over time, which cannot be observed directly on regional scales. Identifying these sources and sinks spatially and temporally is an ill-posed inverse problem, often called a trace-gas flux-inversion problem, whose solution requires the use of both an atmospheric transport model and a spatio-temporal model for the fluxes (e.g., Enting, 2002). In this paper, we present version 1.0 of a global flux-inversion system for the solution of this problem, which we call the WOllongong Methodology for Bayesian Assimilation of Trace-gases (WOMBAT).

A global trace-gas flux-inversion system is designed to infer fluxes from observational data, which are generally available either as point-referenced (flask or in situ) measurements or column-averaged remote-sensing retrievals. The underlying model in an inversion system is usually a state-space model, where the fluxes, or a reduced representation thereof, are the latent states that need to be inferred from data via the use of an atmospheric chemical transport model (CTM). Computationally, flux estimation is done within a standard optimisation framework (e.g., Chevallier et al., 2005; Baker et al., 2006), either via full Bayesian synthesis (Enting 2002, Chapter 3; Mukherjee et al. 2011; Schuh et al. 2019) or via ensemble Kalman filtering (e.g., Peters et al., 2005; Feng et al., 2009). Inversion systems rely, to various extents, on realistic bottom-up estimates of fluxes for the elicitation of an informative prior distribution; an accurate CTM that provides the link between the fluxes and the observed mole fraction; high-quality unbiased measurements; and reliable uncertainty measures on each model component.

The complexity of all modelled processes, from fluxes right up to satellite retrieval errors, inevitably leads to model misspecification (e.g., Engelen et al., 2002). The main causes of misspecification are (i) flux-process dimension-reduction error (e.g., Kaminski et al., 2001), which is a consequence of using a spatio-temporal model for the flux field that is low-dimensional and inflexible; (ii) an inaccurate prior flux mean, variance, and covariance (e.g., Philip et al., 2019); (iii) transport-model errors (e.g., Houweling et al., 2010; Basu et al., 2018; Schuh et al., 2019) arising from the underlying assumed physics, meteorology, and discretisation schemes used (e.g., Lauvaux et al., 2019; McNorton et al., 2020); (iv) retrieval biases (e.g., O'Dell et al., 2018) and incorrect associated measurement-error statistics (e.g., Worden et al., 2017); and (v) measurement-error spatio-temporal correlations that are not fully accounted for (e.g., Chevallier, 2007; Ciais et al., 2010). Two other causes of model misspecification worth noting are an incorrectly specified initial global mole-fraction field, and flux components assumed known in the inversion (i.e., assumed degenerate at their prior mean), such as anthropogenic emissions (e.g., Feng et al., 2019). The latter can be seen as a special case of (i) above, while the effect of the former can generally be minimised by using a realistic initial condition (e.g., Basu et al., 2013) and incorporating a burn-in (or spin-up) period, in which early flux estimates are discarded.

In Sect. 2, we present the underlying statistical framework of WOMBAT, which addresses the implications of model misspecification in four ways: first, by using prior distributions to encode uncertainty over prior beliefs on the fluxes (e.g., Ganesan et al., 2014; Zammit-Mangion et al., 2016); second, by adding a spatio-temporally correlated component of variability to the mole-fraction data model to address some of this (typically unmodelled) correlated model–data discrepancy (Brynjarsdóttir and O'Hagan, 2014); third, by explicitly modelling biases in the mole-fraction data model (generalising the approach of Basu et al., 2013); and fourth, by propagating uncertainty on all unknowns within a fully Bayesian statistical framework wherein inference is made using Markov chain Monte Carlo (MCMC). We note that while the benefits of MCMC are becoming in-





creasingly apparent in *regional* trace-gas inversions (e.g., Mukherjee et al., 2011; Ganesan et al., 2014; Miller et al., 2014;
Zammit-Mangion et al., 2016), its use is still the exception rather than the rule in *global* trace-gas inversions. The use of a
spatio-temporally correlated component of variability leads to computational challenges, which are addressed in Sect. 3.

The fully Bayesian nature of the model, coupled with the introduction of a correlated process in modelling the mole-fraction
field, leads to computational challenges. Section 3 details how we deal with these by defining a specific type of stochastic
process on the irregularly located spatio-temporal errors, one that leads to a sparse precision matrix (Vecchia, 1988; Datta
et al., 2016). Details on how this facilitates the MCMC scheme we implement are deferred to Appendix A. Section 4 discusses
the experimental setup used for running, validating, and implementing WOMBAT on the satellite data analysed in this article.
In Sect. 5, we first conduct an observing system simulation experiment (OSSE) in which the true fluxes are assumed known.
Results from the OSSE demonstrate WOMBAT's validity and also illustrate the importance of modelling biases and correlated
error terms when these are indeed present in the data. We then use WOMBAT to perform flux inversion using the Orbiting
Carbon Observatory-2 (OCO-2) Version 7 retrospective (V7r) dataset as used in the model intercomparison project (MIP) of
Crowell et al. (2019). Our model fitting reveals that about 80% of the total error variance associated with the OCO-2 data used
for the MIP can be explained with a correlated model–data discrepancy term. WOMBAT accounts for this, which results in
posterior distributions over the fluxes that, for the most part, corroborate the results from the ensemble inversions, both on a
regional and on a global scale. In Sect. 5, we also show the utility of WOMBAT in carrying out online bias correction. Section 6
summarises the features of WOMBAT and discusses avenues for future work.

## 2  Bayesian hierarchical model for global flux inversion of trace gases

In this section we outline the spatio-temporal Bayesian hierarchical statistical model (e.g.,  Wikle et al., 2019, Sect. 1.3) that
WOMBAT uses for global flux inversion. The model consists of four sub-models: (1) a flux process model, (2) a mole-fraction
process model, (3) a mole-fraction data model, and (4) a parameter model.

### 2.1  The flux process model

Let $Y_1^0(\mathbf{s}, t)$ denote the prior mean of the trace-gas surface flux at spatial location $\mathbf{s} \in \mathbb{S}^2$ and time $t \in \mathcal{T}$, where $\mathbb{S}^2$ is the surface
of Earth, and $\mathcal{T} \equiv [t_0, t_f]$ is some time interval of interest. The field $Y_1^0(\cdot, \cdot)$ could be set to zero everywhere (e.g., Michalak
et al., 2004) or could be constructed using bottom-up estimates of biospheric and/or anthropogenic fluxes (e.g., Basu et al.,
2013).

In the same vein as conventional Bayesian-synthesis frameworks (e.g., Enting, 2002), we model the true flux, $Y_1(\cdot, \cdot)$, as
$Y_1^0(\cdot, \cdot)$ plus a spatio-temporal field constructed through a sum of $r$ spatio-temporal basis functions. These basis functions
could be space-time step functions, as typically found in variational inversion systems (e.g., Chevallier et al., 2005), discretised
flux "patterns" (e.g., Fan et al., 1998), or they could be a general-purpose basis such as a Fourier basis (e.g., Crowell et al.,
2019, Appendix A4).





Denote the set of pre-specified flux basis functions as $\{\phi_j(\cdot,\cdot) : j = 1,\ldots,r\}$. Our flux process model is a spatio-temporal stochastic process given by,

$$Y_1(\mathbf{s},t) = Y_1^0(\mathbf{s},t) + \sum_{j=1}^{r} \phi_j(\mathbf{s},t)\alpha_j, \quad \mathbf{s} \in \mathbb{S}^2, t \in \mathcal{T}, \tag{1}$$

where the scaling factors $\{\alpha_j : j = 1,\ldots,r\}$ are unknown, are assigned a multivariate probability distribution, and need to be inferred in the inversion framework. Since we assume that $\mathrm{E}(Y_1(\cdot,\cdot)) = Y_1^0(\cdot,\cdot)$, we let $\mathrm{E}(\alpha_j) = 0$ for $j = 1,\ldots,r$. For a given

set of basis functions, the prior belief on the covariance structure of $Y_1(\cdot,\cdot)$ is fully determined by that on $\boldsymbol{\alpha} \equiv (\alpha_1,\ldots,\alpha_r)'$. When the flux is modelled on a space-time grid and space-time step functions are used as basis functions, a prior distribution on $\boldsymbol{\alpha}$ that correlates the flux a priori in space and time is natural (Michalak et al., 2004; Chevallier et al., 2007; Basu et al., 2018). On the other hand, when using large spatial flux "patterns" that have temporally-limited scope, it is generally reasonable to assume that the $\alpha_j$'s corresponding to basis functions in different spatial regions are uncorrelated, but that those associated

with the same spatial region are temporally correlated.

Irrespective of the choice of basis functions, in WOMBAT one expresses prior judgement on $\boldsymbol{\alpha}$ through the model $\boldsymbol{\alpha} \sim \mathrm{Gau}(\mathbf{0},\boldsymbol{\Sigma}_\alpha)$, where $\mathrm{Gau}(\boldsymbol{\mu},\boldsymbol{\Sigma})$ is a Gaussian probability density function of a random vector with mean $\boldsymbol{\mu}$ and covariance matrix $\boldsymbol{\Sigma}$. The covariance matrix $\boldsymbol{\Sigma}_\alpha$ is parameterised through a parameter vector $\boldsymbol{\theta}_\alpha$, which typically contains variances and spatio-temporal length scales in the covariances. Expert elicitation can be used to construct prior distributions on these

parameters too; we describe possible prior distributions when discussing the parameter model in the Bayesian hierarchical model in Sect. 2.4.

The flux model of Eq. (1) may be improved by introducing a dimension-reduction error, also known as aggregation error, on the right-hand side. This error accounts for the fact that the structured basis functions typically span a small (function) space and that, therefore, they cannot reproduce fluxes perfectly. However, since we are unable to deconvolve dimension-reduction error

from other sources of error (e.g., transport-model error) in our mole-fraction data, we model the spatio-temporal variability it introduces collectively with other sources of error (Sect. 2.2).

## 2.2 The mole-fraction process model

Denote the true mole-fraction process at space-height-time location $(\mathbf{s},h,t)$, as $Y_2(\mathbf{s},h,t)$. We only model the mole fraction within our time interval of interest $\mathcal{T}$, starting at time $t_0$, and therefore we express the true mole-fraction field within $\mathcal{T}$ as a

function of the initial mole-fraction process at $t_0$ and the exogenous flux-process inputs in $\mathcal{T}$. Specifically, define $\mathcal{T}_t \equiv [t_0,t]$, for $t_0 \leq t \leq t_f$, as the set of all time points in $\mathcal{T}$ up to and including $t$. The field $Y_2(\cdot,\cdot,\cdot)$ is related to the flux process through the relationship,

$$Y_2(\mathbf{s},h,t) = \mathcal{H}(Y_2(\cdot,\cdot,t_0),Y_1(\cdot,\mathcal{T}_t);\mathbf{s},h,t), \quad \mathbf{s} \in \mathbb{S}^2,\ h \in \mathbb{R}^+,\ t \in \mathcal{T}, \tag{2}$$

where $Y_2(\cdot,\cdot,t_0)$ is the mole-fraction field at time $t_0$, $Y_1(\cdot,\mathcal{T}_t)$ is the flux field evolving over the whole time period $\mathcal{T}_t$, and $\mathcal{H}$

is an operator that solves the underlying chemical transport equations (that are approximately linear for long-lived species such





as $CO_2$; see, e.g., Enting, 2002, Chapter 2). In practice, $\mathcal{H}$ is not known perfectly, but we usually have at hand a reasonable approximation to it, $\hat{\mathcal{H}}$, which is often referred to as the chemical transport model (CTM) or simply as the transport model. Similarly, we will usually have a reasonable approximation to $Y_2(\cdot,\cdot,t_0)$, which we call $\hat{Y}_2(\cdot,\cdot,t_0)$. The use of $\hat{Y}_2(\cdot,\cdot,t_0)$ instead of $Y_2(\cdot,\cdot,t_0)$, and of $\hat{\mathcal{H}}$ instead of $\mathcal{H}$, leads to a residual term $v_2(\cdot,\cdot,\cdot)$ that will inevitably be spatio-temporally
correlated (Enting, 2002, Chapter 9). In particular,

$$Y_2(\mathbf{s},h,t) = \hat{\mathcal{H}}(\hat{Y}_2(\cdot,\cdot,t_0), Y_1(\cdot,\mathcal{T}_t); \mathbf{s},h,t) + v_2(\mathbf{s},h,t), \quad \mathbf{s} \in \mathbb{S}^2,\ h \in \mathbb{R}^+,\ t \in \mathcal{T}, \tag{3}$$

where $v_2(\cdot,\cdot,\cdot)$ is the residual mole-fraction process arising from the use of an approximate initial mole-fraction field, imperfect meteorology inside the transport model, imperfect transport-model parameters and physics, and potentially sub-grid-scale variation in the mole-fraction field when $\hat{\mathcal{H}}$ is a numerical model evaluated at a coarse resolution. It is difficult to place
prior beliefs on the structure of $v_2(\cdot,\cdot,\cdot)$, which we model as statistical error, but it is known that using the approximation $\hat{\mathcal{H}}$ introduces errors that could span hundreds of kilometres and several days (Lauvaux et al., 2019; McNorton et al., 2020). Transport-model implementations tend to differ considerably in their vertical and inter-hemispheric mixing behaviour, and flux-inversion estimates are known to be particularly sensitive to transport-model choice (Gurney et al., 2002; Schuh et al., 2019). Note that $v_2(\cdot,\cdot,\cdot)$ is also likely to depend on $Y_1(\cdot,\cdot)$, and that we ignore this dependence for model simplicity in what
follows.

The assumed linear behaviour of the underlying dynamics for $CO_2$ is important. First, it allows us to model the effect of the approximate initial mole-fraction field, $\hat{Y}_2(\cdot,\cdot,t_0)$, separately from that of the fluxes (e.g., Enting, 2002, Chapter 10), so that Eq. (3) is of the form

$$Y_2(\mathbf{s},h,t) = \hat{\mathcal{H}}(\hat{Y}_2(\cdot,\cdot,t_0), 0; \mathbf{s},h,t) + \hat{\mathcal{H}}(0, Y_1(\cdot,\mathcal{T}_t); \mathbf{s},h,t) + v_2(\mathbf{s},h,t), \tag{4}$$

for $\mathbf{s} \in \mathbb{S}^2$, $h \in \mathbb{R}^+$, and $t \in \mathcal{T}$. Second, it allows us to express the mole-fraction field as a linear combination of the individual responses from the basis functions used to construct $Y_1(\cdot,\cdot)$, as we now show. Substituting Eq. (1) into Eq. (4), we have that

$$Y_2(\mathbf{s},h,t) = Y_2^0(\mathbf{s},h,t) + \sum_{j=1}^{r} \hat{\psi}_j(\mathbf{s},h,t)\alpha_j + v_2(\mathbf{s},h,t), \tag{5}$$

for $\mathbf{s} \in \mathbb{S}^2$, $h \in \mathbb{R}^+$, and $t \in \mathcal{T}$, where $Y_2^0(\mathbf{s},h,t) \equiv \hat{\mathcal{H}}(\hat{Y}_2(\cdot,\cdot,t_0), 0; \mathbf{s},h,t) + \hat{\mathcal{H}}(0, Y_1^0(\cdot,\mathcal{T}_t); \mathbf{s},h,t)$; $\hat{\psi}_j(\mathbf{s},h,t) \equiv \hat{\mathcal{H}}(0, \phi_j(\cdot,\mathcal{T}_t); \mathbf{s},h,t)$, for $j = 1,\ldots,r$, are basis functions in mole-fraction space, often termed *response functions* (e.g.,
Saeki et al., 2013); and $v_2(\mathbf{s},h,t)$ is the residual term given in Eq. (3). We assume that $\mathrm{E}(v_2(\cdot,\cdot,\cdot)) = 0$, so that $Y_2^0(\mathbf{s},h,t)$ can be seen as the prior expectation of the mole-fraction field at $(\mathbf{s},h,t)$ under $\hat{\mathcal{H}}$ and $\hat{Y}_2(\cdot,\cdot,t_0)$. That is, it is the mole-fraction field generated by running our CTM with the input fluxes set to the prior expected flux, and with the mole-fraction field at $t_0$ set to $\hat{Y}_2(\cdot,\cdot,t_0)$.

### 2.3 The mole-fraction data model

Fluxes cannot be observed directly at the spatial and temporal scales of interest. Flux inversion therefore proceeds by "constraining" the flux field using column-averaged retrievals or point-referenced measurements of mole fraction. We use $Z_{2,i}$ to





denote the $i$th mole-fraction measurement or retrieval, where $i \in \{1, \ldots, m\}$ indexes the datum used in the inversion, and $m$ is the number of data used in the inversion.

Point-referenced (PR) measurements of mole fraction are generally made at or near Earth's surface, using instruments on
towers or in aircraft. The mole-fraction data model for these measurements is therefore given by

$$Z_{2,i} = Y_2(\mathbf{s}_i, h_i, t_i) + \epsilon_i, \quad \text{if } Z_{2,i} \text{ is from a point-referenced instrument,} \tag{6}$$

where $Z_{2,i}$ is the observed mole fraction at $(\mathbf{s}_i, h_i, t_i)$, and $\epsilon_i$ is mean-zero Gaussian measurement error with a model for its variance parameter presented below in Sect. 2.4. Measurement errors associated with point-referenced instruments are generally small and (usually) not correlated in space and time. Despite this, such data are not immune to the effects of spatio-
temporal correlations induced by the CTM in the process model, and they may even be more susceptible than column-averaged retrievals due to the combined effect of their usual proximity to the surface and the discretisations employed when simulating approximate transport (Rayner and O'Brien, 2001; Basu et al., 2018).

Column-averaged (CA) retrievals, such as XCO$_2$ (where "X" refers to the column averaged nature of the retrievals) from the OCO-2 satellite or the TCCON sites, are noisier than PR measurements, although TCCON less so. In particular, since the raw
spectral information collected for the retrieval is affected by environmental factors such as aerosols (O'Dell et al., 2012), the errors can contain biases as well as exhibit spatio-temporal correlations. These biases can also be instrument-mode dependent (e.g., land glint [LG] vs land nadir [LN] retrievals for OCO-2; see Sect. 4.4.1). The vertical averaging operation also involves an averaging kernel and an a priori bias correction, which are both specific to the retrieval, and which arise from the algorithm used for the retrieval. In general, this relationship can be expressed as

$$Z_{2,i} = \hat{\mathcal{A}}_i(Y_2(\mathbf{s}_i, \cdot, t_i)) + b_i + v_{Z_{2,i}} + \epsilon_i, \quad \text{if } Z_{2,i} \text{ is a column-averaged retrieval,} \tag{7}$$

where $(\mathbf{s}_i, t_i)$ is the space-time location of the retrieval, $\hat{\mathcal{A}}_i$ is the assumed (but necessarily approximate) observation operator of the $i$th retrieval that column-averages the mole fraction field via an averaging kernel; $b_i$ is bias; $v_{Z_{2,i}}$ is mean-zero spatio-temporally correlated random error; and $\epsilon_i$ is mean-zero uncorrelated random error. The bias and error terms arise from the use of an approximate observation operator. Surface-based or remotely sensed CA retrievals are sometimes provided as "bias-
corrected". In this case, the data model for these retrievals is identical to Eq. (7), but with the bias component omitted.

Substituting Eq. (5) into Eq. (6) and Eq. (7) we see that, in general, we have that

$$Z_{2,i} = \hat{Z}_{2,i}^0 + \sum_{j=1}^{r} \hat{\psi}_{j,i} \alpha_j + b_i + \xi_i + \epsilon_i, \tag{8}$$

where, for a PR measurement, $\hat{Z}_{2,i}^0 \equiv Y_2^0(\mathbf{s}_i, h_i, t_i)$, $\hat{\psi}_{j,i} \equiv \hat{\psi}_j(\mathbf{s}_i, h_i, t_i)$, for $j = 1, \ldots, r$, $b_i = 0$; and $\xi_i \equiv v_2(\mathbf{s}_i, h_i, t_i)$; while for a CA retrieval, $\hat{Z}_{2,i}^0 \equiv \hat{\mathcal{A}}_i(Y_2^0(\mathbf{s}_i, \cdot, t_i))$, $\hat{\psi}_{j,i} \equiv \hat{\mathcal{A}}_i(\hat{\psi}_j(\mathbf{s}_i, \cdot, t_i))$, for $j = 1, \ldots, r$; and $\xi_i \equiv \hat{\mathcal{A}}_i(v_2(\mathbf{s}_i, \cdot, t_i)) + v_{Z_{2,i}}$. Bias-
corrected retrievals are given by Eq. (8) but with the bias component $b_i$ omitted. Note that, for identifiability reasons, we have modelled all the possibly correlated error terms using one component, $\{\xi_i\}$.

Flux inversions can make use of both PR measurements and CA retrievals simultaneously, and hence it is convenient to provide a data model that encapsulates both types of measurements. It is also often the case that measurements from the same



instrument type can be divided into groups that can be expected to have similar characteristics, such as group-specific bias and
error properties. A given group could contain, for example, PR data from the same in situ instrument, or CA retrievals from
a particular remote sensing instrument under a specific retrieval mode (e.g., land nadir). Hence, we consider the following
general data model, where different groups have different terms, but the overall structure is the same:

$$\mathbf{Z}_{2,g} = \hat{\mathbf{Z}}_{2,g}^0 + \hat{\mathbf{\Psi}}_g \boldsymbol{\alpha} + \mathbf{b}_g + \boldsymbol{\xi}_g + \boldsymbol{\epsilon}_g, \quad g = 1, \dots, n_g, \tag{9}$$

where $n_g$ is the number of groups; $\mathbf{Z}_{2,g}$ contains the data in group $g$; $\hat{\mathbf{Z}}_{2,g}^0$ are the prior expected mole fractions in group $g$
under the approximate transport model, the approximate mole-fraction field at $t_0$ and, if the groups consist of CA retrievals,
under the approximate observation operators; $\hat{\mathbf{\Psi}}_g$ are the response functions in group $g$ evaluated at either the PR locations
(in the case of PR measurements) or averaged over a column via an approximate observation operator (in the case of CA
retrievals); $\mathbf{b}_g \equiv \mathbf{A}_g \boldsymbol{\beta}_g$ are group-specific biases, with $\mathbf{A}_g$ a group-specific design matrix and $\boldsymbol{\beta}_g$ the corresponding weights;
$\boldsymbol{\xi}_g$ is the group $g$'s vector of correlated errors; and $\boldsymbol{\epsilon}_g$ is the group's vector of uncorrelated errors. When the data in group
$g$ are considered to be unbiased (or already bias-corrected), the term $\mathbf{A}_g \boldsymbol{\beta}_g = \mathbf{0}$. The variables constituting $\boldsymbol{\beta}_g$ and $\boldsymbol{\epsilon}_g$, for
$g = 1, \dots, n_g$, are mutually independent within and across groups.

The correlation between elements of $\boldsymbol{\xi}_g$ associated with measurements that are proximal in space and time is stronger than
between those that are farther apart. Yet, while spatio-temporal correlation in model–data discrepancies is widely acknowledged
(Chevallier, 2007; Ciais et al., 2010; Mukherjee et al., 2011; Miller et al., 2020), the general consensus is that using just the
variance of $\boldsymbol{\xi}_g$ to add to the variance of the uncorrelated component is sufficient (e.g., Michalak et al., 2005; Basu et al., 2013;
Deng et al., 2016). However, as we show in our OSSE in Sect. 5.1, even when a measurement-error variance inflation factor is
estimated, statistical efficiency is lost under the assumption of uncorrelated errors if the errors truly are correlated. The main
reason not to routinely model spatio-temporal correlations in global flux inversion appears to be computational; we discuss a
way to rectify this bottleneck in Sect. 3.

## 2.4 The parameter model

The parameter model (i.e., prior distributions on parameters) is dependent on the specification of the flux process model, the
mole-fraction process model, and the mole-fraction data model. Here, we describe the parameter model we use in the OSSE
and in the MIP comparison in Sect. 5.

**Parameters of the flux process model:** In the experiments given below in Sect. 5, our flux basis functions are from bottom-
up inventories that are divided into $r_s$ spatial regions and $r_t$ time spans. This construction yields $r = r_s \times r_t$ basis functions,
and it naturally suggests a temporal partitioning of $\boldsymbol{\alpha}$ into $(\boldsymbol{\alpha}_1', \dots, \boldsymbol{\alpha}_{r_t}')'$, where each $\boldsymbol{\alpha}_k \in \mathbb{R}^{r_s}$, for $k = 1, \dots, r_t$. This, in
turn, suggests that a suitable model for $\boldsymbol{\alpha}$ is the vector-autoregressive process, similar to that used by Peters et al. (2005)
and Dahlén et al. (2020). Specifically, $\boldsymbol{\alpha}_{k+1} = \mathbf{M}(\boldsymbol{\kappa})\boldsymbol{\alpha}_k + \mathbf{w}_k$, for $k = 1, \dots, r_t$, where, in our examples, we constrain the
matrix $\mathbf{M}(\boldsymbol{\kappa})$ to be diagonal with non-zero elements equal to $\boldsymbol{\kappa} \equiv (\kappa_1, \dots, \kappa_{r_s})'$; and we let $\mathbf{w}_k \sim \text{Gau}(\mathbf{0}, \boldsymbol{\Sigma}_w)$, where the
precision matrix $\mathbf{Q}_w \equiv \boldsymbol{\Sigma}_w^{-1}$ is diagonal with positive elements $\boldsymbol{\tau}_w \equiv (\tau_{w,j} : j = 1, \dots, r_s)'$. The flux-process parameters are
therefore $\boldsymbol{\theta}_\alpha \equiv (\boldsymbol{\kappa}', \boldsymbol{\tau}_w')'$, which in turn govern the covariance matrix of $\boldsymbol{\alpha}$, notated as $\boldsymbol{\Sigma}_\alpha \equiv \text{var}(\boldsymbol{\alpha})$. There is an ordering of





$\boldsymbol{\alpha}$ for which $\boldsymbol{\Sigma}_\alpha^{-1}$ is block diagonal with each block a tridiagonal matrix (see Appendix A). Either sequential estimation (e.g., Kalman filtering/smoothing) or batch Bayesian updating can be used to make inference on $\boldsymbol{\alpha}$. In our case we use the latter, and we take advantage of efficient algorithms that are available for sparse-linear-algebraic computations.

We expect that $\{\boldsymbol{\alpha}_k\}$ are positively correlated in time. Therefore, for the prior distributions for $\{\kappa_j\}$, we use the beta distribution, which has support on the interval $[0,1]$: Independently, $\kappa_j \sim \mathrm{Beta}(a_{\kappa,j}, b_{\kappa,j})$, for $j = 1, \ldots, r_s$, where $\{a_{\kappa,j}\}$ and $\{b_{\kappa,j}\}$ are fixed and assumed known. For prior distributions on the precision parameters $\{\tau_{w,j}\}$, we use gamma distributions with shape parameters $\{\nu_{w,j}\}$ and rate parameters $\{\omega_{w,j}\}$, which are fixed and assumed known: Independently, $\tau_{w,j} \sim \mathrm{Ga}(\nu_{w,j}, \omega_{w,j})$, for $j = 1, \ldots, r_s$.

Michalak et al. (2005) suggested that variance parameters could be estimated directly in a maximum-likelihood framework. The use of prior distributions on $\boldsymbol{\kappa}$ and $\boldsymbol{\tau}_w$ adds an extra level of flexibility and allows the modeller to express the "uncertainty on the uncertainties" in an inversion framework (e.g., Ganesan et al., 2014). One can configure these prior distributions to be extremely informative and, effectively, fix the prior model for the flux, or else make them extremely uninformative so that the posterior maximum-a-posteriori (MAP) estimates of the parameters in the prior flux-process model converge to the maximum-

likelihood estimates. This choice can be made on a region-by-region basis, as is the case in our experiments (Sect. 4.5), where land regions are given uninformative priors, and ocean regions are given informative ones.

**CA mole-fraction retrieval bias parameters:** In Sect. 5, we consider OCO-2 retrievals and a different set of bias parameters for each instrument mode. In this context, $\boldsymbol{\beta}_g$ is associated with a particular instrument mode and a single element of $\boldsymbol{\beta}_g$ captures the bias arising from, for example, aerosol presence. Experiments (see Sect. 5.3) reveal that these bias parameters

are quite easily constrained in an inversion framework. When constructing the model for each $\boldsymbol{\beta}_g$, we first standardise each row in $\mathbf{A}_g$ so that the covariates have unit marginal empirical variance. Then we model $\{\boldsymbol{\beta}_g\}$ as follows: Independently, $\boldsymbol{\beta}_g \sim \mathrm{Gau}(\mathbf{0}, \sigma_\beta^2 \mathbf{I})$, for $g = 1, \ldots, n_g$, with $\sigma_\beta^2 = 100$ and where $\mathbf{I}$ is the identity matrix. This choice for $\sigma_\beta^2$ renders the prior distribution uninformative for the data-set sizes we consider in our experiments.

**Model–data discrepancy and measurement-error parameters:** The retrievals used to perform inversions often come with

prescribed variances, $\mathbf{v}_g^{\mathrm{ps}}$, that account for both retrieval error, correlated or otherwise, and CTM error. For example, the MIP protocol of Crowell et al. (2019) prespecified these variances. We therefore let the *total marginal variance* of $\boldsymbol{\xi}_g + \boldsymbol{\epsilon}_g$ be equal to $\gamma_g \cdot \mathbf{v}_g^{\mathrm{ps}}$, where the inflation-factor parameter $\gamma_g$ accounts for the possibility of misspecified variances (Worden et al., 2017). To deconvolve $\boldsymbol{\xi}_g$ and $\boldsymbol{\epsilon}_g$, we first construct the correlation matrix $\mathbf{R}_{\xi_g} \equiv \mathrm{corr}(\boldsymbol{\xi}_g)$ using a spatio-temporal correlation function $R_{\xi_g}(\cdot, \cdot; \boldsymbol{\ell}_{\xi_g})$, where $\boldsymbol{\ell}_{\xi_g}$ are length scales that need to be inferred from data. We then enforce the total marginal

variance constraint by defining the covariance matrices of $\boldsymbol{\xi}_g$ and $\boldsymbol{\epsilon}_g$ to be $\boldsymbol{\Sigma}_{\xi_g} \equiv \mathrm{var}(\boldsymbol{\xi}_g) = \mathrm{diag}(\rho_g \gamma_g \cdot \mathbf{v}_g^{\mathrm{ps}})^{1/2} \mathbf{R}_{\xi_g} \mathrm{diag}(\rho_g \gamma_g \cdot \mathbf{v}_g^{\mathrm{ps}})^{1/2}$ and $\boldsymbol{\Sigma}_{\epsilon_g} \equiv \mathrm{var}(\boldsymbol{\epsilon}_g) = \mathrm{diag}((1 - \rho_g) \gamma_g \cdot \mathbf{v}_g^{\mathrm{ps}})$. The parameter $\rho_g$ represents the relative contribution of the correlated-error variance (comprising both CTM error and, if present, correlated measurement error) to the total inflated prescribed variance.

    We model the inflation factors $\{\gamma_g\}$ using inverse-gamma distributions: Independently, $\gamma_g \sim \mathrm{IG}(\nu_{\gamma_g}, \omega_{\gamma_g})$, $\quad g = 1, \ldots, n_g$, where the shape and rate parameters, $\{\nu_{\gamma_g}\}$ and $\{\omega_{\gamma_g}\}$, are fixed and assumed known. We model the relative-contribution fac-

tors $\{\rho_g\}$ using standard uniform distributions: Independently, $\rho_g \sim \mathrm{Unif}(0,1)$, for $g = 1, \ldots, n_g$. We use gamma prior distributions to model the length scales $\{\boldsymbol{\ell}_g\}$ in the correlation function: Independently, $\ell_{g,j} \sim \mathrm{Ga}(\nu_{\ell_g,j}, \omega_{\ell_g,j})$, for $j = 1, \ldots, n_{\ell_g}$; $g =$





$1, \ldots, n_g$. We collect together the unknown parameters that determine the variances and covariances of the correlated component of the error into $\boldsymbol{\theta}_{\xi_g} \equiv (\rho_g, \boldsymbol{\ell}_g')'$, for $g = 1, \ldots, n_g$.

## 2.5 Summary of the Bayesian hierarchical model and computation

The Bayesian hierarchical model, which we use in Sect. 5, can be written succinctly as follows: Independently,

$$
\begin{aligned}
\text{Flux autoregressive parameters:} \quad & \kappa_j \sim \text{Beta}(a_{\kappa,j}, b_{\kappa,j}), \quad j = 1, \ldots, r_s, \\
\text{Flux innovation precisions:} \quad & \tau_{w,j} \sim \text{Ga}(\nu_{w,j}, \omega_{w,j}), \quad j = 1, \ldots, r_s, \\
\text{Flux scaling factors:} \quad & \boldsymbol{\alpha} \mid \boldsymbol{\kappa}, \boldsymbol{\tau}_w \sim \text{Gau}(\mathbf{0}, \boldsymbol{\Sigma}_\alpha), \\
\text{Measurement-bias coefficients:} \quad & \boldsymbol{\beta}_g \sim \text{Gau}(\mathbf{0}, \sigma_\beta^2 \mathbf{I}), \quad g = 1, \ldots, n_g, \\
\text{Error variance inflation factors:} \quad & \gamma_g \sim \text{IG}(\nu_{\gamma_g}, \omega_{\gamma_g}), \quad g = 1, \ldots, n_g, \\
\text{Error length scales:} \quad & \ell_{g,j} \sim \text{Ga}(\nu_{\ell_g,j}, \omega_{\ell_g,j}), \quad j = 1, \ldots, n_{\ell_g}; \ g = 1, \ldots, n_g, \\
\text{Error proportion:} \quad & \rho_g \sim \text{Unif}(0,1), \quad g = 1, \ldots, n_g, \\
\text{Likelihood:} \quad & \mathbf{Z}_2 \mid \boldsymbol{\beta}, \boldsymbol{\alpha}, \boldsymbol{\theta}_{\xi,\epsilon} \sim \text{Gau}(\hat{\mathbf{Z}}_2^0 + \hat{\boldsymbol{\Psi}}\boldsymbol{\alpha} + \mathbf{A}\boldsymbol{\beta}, \boldsymbol{\Sigma}_\xi + \boldsymbol{\Sigma}_\epsilon),
\end{aligned}
$$

where $\mathbf{Z}_2 \equiv (\mathbf{Z}_{2,1}', \ldots, \mathbf{Z}_{2,n_g}')'$, $\hat{\mathbf{Z}}_2^0 \equiv (\hat{\mathbf{Z}}_{2,1}^{0'}, \ldots, \hat{\mathbf{Z}}_{2,n_g}^{0'})'$, $\hat{\boldsymbol{\Psi}} \equiv (\hat{\boldsymbol{\Psi}}_1', \ldots, \hat{\boldsymbol{\Psi}}_{n_g}')'$, $\mathbf{A} \equiv \text{bdiag}(\mathbf{A}_1, \ldots, \mathbf{A}_{n_g})$, $\boldsymbol{\beta} \equiv (\boldsymbol{\beta}_1', \ldots, \boldsymbol{\beta}_{n_g}')'$,

$\boldsymbol{\Sigma}_\xi \equiv \text{bdiag}(\boldsymbol{\Sigma}_{\xi_1}, \ldots, \boldsymbol{\Sigma}_{\xi_{n_g}})$, $\boldsymbol{\Sigma}_\epsilon \equiv \text{bdiag}(\boldsymbol{\Sigma}_{\epsilon_1}, \ldots, \boldsymbol{\Sigma}_{\epsilon_{n_g}})$, and $\boldsymbol{\theta}_{\xi,\epsilon} \equiv ((\boldsymbol{\theta}_{\xi_g}', \gamma_g) : g = 1, \ldots, n_g)'$. Here, $\text{bdiag}(\cdot)$ constructs a block-diagonal matrix from its arguments. A graphical model summarising the relationships between the variables is given in Fig. 1.

The joint posterior distribution over all quantities can be estimated using a Gibbs sampler, which successively "updates" parameters using their full conditional distributions. When the conditional distributions cannot be sampled from directly (in

particular, for the parameters $\boldsymbol{\kappa}, \boldsymbol{\ell}_g$, and $\rho_g$, for $g = 1, \ldots, n_g$), we employ a slice sampler (Neal, 2003) to obtain samples. Details are given in Appendix A.

## 3 The model–data discrepancy term

The posterior distribution over all the unknown parameters in our model is given by

$$
p(\boldsymbol{\alpha}, \boldsymbol{\beta}, \boldsymbol{\kappa}, \boldsymbol{\tau}_w, \boldsymbol{\theta}_{\xi,\epsilon} \mid \mathbf{Z}_2) \propto p(\mathbf{Z}_2 \mid \boldsymbol{\beta}, \boldsymbol{\alpha}, \boldsymbol{\theta}_{\xi,\epsilon}) p(\boldsymbol{\alpha} \mid \boldsymbol{\kappa}, \boldsymbol{\tau}_w) p(\boldsymbol{\beta}) p(\boldsymbol{\theta}_{\xi,\epsilon}) p(\boldsymbol{\kappa}) p(\boldsymbol{\tau}_w). \tag{10}
$$

The log of the first two terms on the right-hand side of Eq. (10) are expressions that are commonly seen in optimisation-based flux-inversion frameworks. In particular, we have that

$$
\begin{aligned}
\log p(\mathbf{Z}_2 \mid \boldsymbol{\beta}, \boldsymbol{\alpha}, \boldsymbol{\theta}_{\xi,\epsilon}) p(\boldsymbol{\alpha} \mid \boldsymbol{\kappa}, \boldsymbol{\tau}_w) = & -\frac{1}{2}\log|\boldsymbol{\Sigma}_\xi + \boldsymbol{\Sigma}_\epsilon| - \frac{1}{2}(\mathbf{Z}_2 - \mathbf{Z}_{2,p}(\boldsymbol{\beta}, \boldsymbol{\alpha}))'(\boldsymbol{\Sigma}_\xi + \boldsymbol{\Sigma}_\epsilon)^{-1}(\mathbf{Z}_2 - \mathbf{Z}_{2,p}(\boldsymbol{\beta}, \boldsymbol{\alpha})) \\
& -\frac{1}{2}\log|\boldsymbol{\Sigma}_\alpha| - \frac{1}{2}(\boldsymbol{\alpha} - \boldsymbol{\alpha}_p)'\boldsymbol{\Sigma}_\alpha^{-1}(\boldsymbol{\alpha} - \boldsymbol{\alpha}_p) + \text{const.},
\end{aligned}
$$





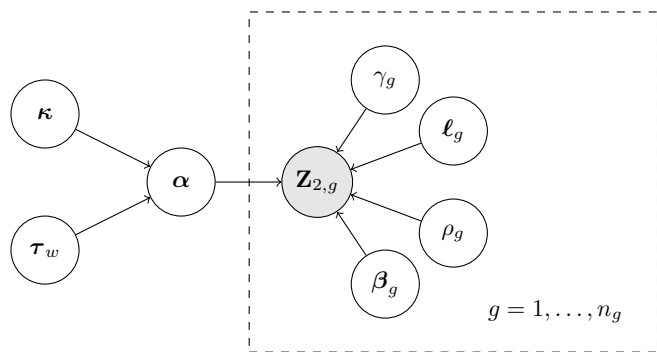

**Figure 1.** Graphical model summarising the relationship between the variables and parameters to be inferred and the grouped data $\{\mathbf{Z}_{2,g} : g = 1, \ldots, n_g\}$.

where $\mathbf{Z}_{2,p}(\boldsymbol{\beta}, \boldsymbol{\alpha}) \equiv \hat{\mathbf{Z}}_2^0 + \hat{\boldsymbol{\Psi}}\boldsymbol{\alpha} + \mathbf{A}\boldsymbol{\beta}$; in our case we set $\boldsymbol{\alpha}_p = \mathbf{0}$; and "const." denotes a constant. The primary differences

between our framework and the usual optimisation-based flux-inversion frameworks are the presence of the log-determinants, which penalise covariance matrices that have large determinants (large correlations and/or large variances), and the presence of off-diagonal elements in the matrix $\boldsymbol{\Sigma}_\xi + \boldsymbol{\Sigma}_\epsilon$. Note that the log-determinants can only be omitted when all covariance matrices are considered known a priori.

The computational complexity of the Gibbs sampler is dominated by that of the log-likelihood function, $\log p(\mathbf{Z}_2 \mid$

$\boldsymbol{\beta}, \boldsymbol{\alpha}, \boldsymbol{\theta}_{\xi,\epsilon})$, which is the sum of the group-wise log-likelihood functions, $\log p(\mathbf{Z}_{2,g} \mid \boldsymbol{\beta}, \boldsymbol{\alpha}, \boldsymbol{\theta}_{\xi,\epsilon})$, for $g = 1, \ldots, n_g$. For computationally efficient inference, we must ensure that each group-wise log-likelihood function is simple to evaluate. From Eq. (9), the group-wise log-likelihood is given by,

$$
\begin{aligned}
\log p(\mathbf{Z}_{2,g} \mid \boldsymbol{\beta}, \boldsymbol{\alpha}, \boldsymbol{\theta}_{\xi,\epsilon}) = & -\frac{m_g}{2} \log 2\pi - \frac{1}{2} \log |\boldsymbol{\Sigma}_g| \\
& -\frac{1}{2} (\mathbf{Z}_{2,g} - \hat{\mathbf{Z}}_{2,g}^0 - \hat{\boldsymbol{\Psi}}_g \boldsymbol{\alpha} - \mathbf{A}_g \boldsymbol{\beta}_g)' \boldsymbol{\Sigma}_g^{-1} (\mathbf{Z}_{2,g} - \hat{\mathbf{Z}}_{2,g}^0 - \hat{\boldsymbol{\Psi}}_g \boldsymbol{\alpha} - \mathbf{A}_g \boldsymbol{\beta}_g),
\end{aligned}
\tag{11}
$$

for $g = 1, \ldots, n_g$, where $\boldsymbol{\Sigma}_g \equiv \boldsymbol{\Sigma}_{\xi_g} + \boldsymbol{\Sigma}_{\epsilon_g}$, and $m_g$ is the number of observations and/or retrievals in group $g$. From a com-

putational perspective there are two components in Eq. (11) that can present difficulties. The first component is the matrix $\hat{\boldsymbol{\Psi}}_g$; this matrix is dense, and its number of elements scales linearly with both the number of data points and the number of basis functions, $r$. Fortunately, this matrix only needs to be evaluated once using the CTM, and it can typically be generated efficiently on a large parallel computing infrastructure. The second component is the matrix $\boldsymbol{\Sigma}_g$, which is of size $m_g \times m_g$ and generally dense. Recall from Sect. 2.4 that this covariance matrix is constructed from $\gamma_g, \ell_{\xi_g}$, and $\rho_g$, which are sampled within

the Gibbs sampler. Therefore, this covariance matrix needs to be re-constructed at each sampler iteration. This is infeasible for the $m_g \approx 100,000$ retrievals used in this study.





We deal with the denseness of $\boldsymbol{\Sigma}_g$ by using an approximation first proposed by Vecchia (1988). Here, one first orders the elements of $\boldsymbol{\xi}_g$. Then one approximates the joint distribution of $\boldsymbol{\xi}_g$ as,

$$p(\boldsymbol{\xi}_g) = p(\xi_{g,1}) \prod_{i=2}^{m_g} p(\xi_{g,i} \mid \boldsymbol{\xi}_{g,1:(i-1)}) \approx p(\xi_{g,1}) \prod_{i=2}^{m_g} p(\xi_{g,i} \mid \boldsymbol{\xi}_{g,\mathcal{N}_{g,i}}), \qquad (12)$$

where $\mathcal{N}_{g,i}$ is the "past" neighbour set of the $i$th datum in group $g$, which contains a (very small) subset of the integers between, and including, 1 and $(i-1)$. It can be shown that this formulation leads to a valid distribution for $\boldsymbol{\xi}_g$ that approximates the true joint distribution. The approximate distribution is Gaussian with mean $\mathbf{0}$ and a sparse precision matrix, $\boldsymbol{\Sigma}_{\xi_g}^{-1}$, with the degree of sparsity closely connected to the sizes of the sets $\{\mathcal{N}_{g,i} : i = 1, \ldots, m_g\}$.

In the version of WOMBAT presented here, we consider a special case of Eq. (12), where the observations are ordered
in time and where the correlation function $R_{\xi_g}(\cdot, \cdot; \ell_{\xi_g})$ is simply an exponential function of temporal separation. In this case, one only needs to consider one (temporal) length-scale parameter per group, $\ell_{g,1}$, for $g = 1, \ldots, n_g$. The motivations for this simplification are twofold. First, the remote-sensing instrument we consider in Sect. 5 flies on a satellite that is in a sun-synchronous orbit, so that correlation in time is a proxy for along-track correlations. This model for characterising correlation in the errors was suggested and used by Chevallier (2007). Second, the use of an exponential correlation function
allows the approximation in Eq. (12) to become an equality, where $\mathcal{N}_{g,i} = i - 1$. This is a manifestation of the so-called "screening effect", where the exponential correlation function induces a first-order conditional-independence structure. Now, $\boldsymbol{\Sigma}_g^{-1} = (\boldsymbol{\Sigma}_{\xi_g} + \boldsymbol{\Sigma}_{\epsilon_g})^{-1}$ and $|\boldsymbol{\Sigma}_g| = |\boldsymbol{\Sigma}_{\xi_g} + \boldsymbol{\Sigma}_{\epsilon_g}|$ , where $\boldsymbol{\Sigma}_{\xi_g}^{-1}$ is very sparse and $\boldsymbol{\Sigma}_{\epsilon_g}^{-1}$ is diagonal. Efficient computations of Eq. (11) therefore follows by expressing $\boldsymbol{\Sigma}_g^{-1}$ and $|\boldsymbol{\Sigma}_g|$ in terms of these sparse matrices. Specifically, $\boldsymbol{\Sigma}_g^{-1}$ and $|\boldsymbol{\Sigma}_g|$ are evaluated through the use of the Sherman–Morrison–Woodbury matrix identity and a matrix-determinant lemma (e.g.,
Henderson and Searle, 1981, and Appendix A).

## 4   Setup of flux-inversion framework

This section gives the setup needed for Sect. 5, where we compare the inversions from WOMBAT v1.0 to those from the OCO-2 MIP (Crowell et al., 2019). In the MIP, inversions followed a predefined protocol; we therefore configured WOMBAT to follow the same protocol. The MIP prescribed the data to be used, including both preprocessed point referenced and remotely
sensed data from OCO-2 between 06 September 2014 and 01 April 2017. Participants were tasked to provide flux estimates for the years 2015 and 2016. The protocol also specified a fossil-fuel flux field that had to be assumed fixed and known, in order to facilitate the interpretation of the differences in flux estimates obtained by the different participants. All other modelling choices (e.g., transport model, prior fluxes) were left to individual participants.





### 4.1 Prior expected flux

Our prior expectation of the spatio-temporal flux process, $Y_1^0(\cdot,\cdot)$, is constructed from inventories of different types of fluxes through the decomposition

$$
Y_1^0(\mathbf{s},\cdot) = \begin{cases} Y_1^{0,\mathrm{ff}}(\mathbf{s},\cdot) + Y_1^{0,\mathrm{bio}}(\mathbf{s},\cdot) + Y_1^{0,\mathrm{bf}}(\mathbf{s},\cdot) + Y_1^{0,\mathrm{fire}}(\mathbf{s},\cdot), & \text{if } \mathbf{s} \text{ is over land,} \\ Y_1^{0,\mathrm{ff}}(\mathbf{s},\cdot) + Y_1^{0,\mathrm{ocn}}(\mathbf{s},\cdot), & \text{if } \mathbf{s} \text{ is over ocean,} \end{cases}
\tag{13}
$$

where $Y_1^{0,\mathrm{ff}}(\cdot,\cdot)$ corresponds to fossil fuel emissions, $Y_1^{0,\mathrm{bio}}(\cdot,\cdot)$ to terrestrial biospheric fluxes, $Y_1^{0,\mathrm{bf}}(\cdot,\cdot)$ to biofuel emissions, $Y_1^{0,\mathrm{fire}}(\cdot,\cdot)$ to fire emissions, and $Y_1^{0,\mathrm{ocn}}(\cdot,\cdot)$ to ocean-air exchange fluxes. We now describe these components in more detail:

– Fossil-fuel emissions: For $Y_1^{0,\mathrm{ff}}(\cdot,\cdot)$ we use the Open-source Data Inventory for Anthropogenic $CO_2$ monthly fossil-fuel emissions (ODIAC2016; Oda and Maksyutov, 2011; Oda et al., 2018) with Temporal Improvements for Modeling Emissions by Scaling (TIMES) weekly scaling factors (Nassar et al., 2013). This term also includes emissions from international aviation and shipping. The fossil-fuel fluxes are prescribed by the MIP protocol.

– Biospheric flux: This flux is a result of the interaction between the atmosphere and trees, shrubs, grasses, soils, dead
wood, leaf litter, and other biota. It is defined by the formula $GPP - R_A - R_H$, where GPP stands for gross primary production and represents the uptake of carbon by plants due to photosynthesis; $R_A$ is autotrophic respiration, the release of carbon through respiration by plants; and $R_H$ is heterotrophic respiration, the release of carbon through the metabolic action of bacteria, fungi, and animals. For $Y_1^{0,\mathrm{bio}}(\cdot,\cdot)$ we use one of the two priors used in CarbonTracker 2019 (Jacobson et al., 2020), specifically that based on the Carnegie–Ames–Stanford Approach (CASA) biogeochemical model (Potter
et al., 1993; Giglio et al., 2013).

– Biofuel emissions: These emissions result from the burning of wood, charcoal, and agricultural waste for energy, as well as the burning of agricultural fields. For $Y_1^{0,\mathrm{bf}}(\cdot,\cdot)$ we use the estimates of Yevich and Logan (2003) that in turn were based on data from 1985.

– Fire emissions: These emissions correspond to those from vegetative fires (wildfires), which may start either naturally
or by humans. For $Y_1^{0,\mathrm{fire}}(\cdot,\cdot)$ we use the Quick Fire Emissions Dataset, version 2 (QFED2; Darmenov and da Silva, 2015).

– Ocean–air exchange: These fluxes are a result of ocean–air differences in partial pressure of $CO_2$. For $Y_1^{0,\mathrm{ocn}}(\cdot,\cdot)$ we use the estimates of Takahashi et al. (2002), with annual scalings reflecting increasing uptake of $CO_2$ as described by Takahashi et al. (2009).

A summary of these components is provided in Table 1.




**Table 1.** Components of the flux prior $Y_1^0(\mathbf{s}, t)$ used in Eq. (13).

| Symbol | Type | Inventory name | Resolution | Reference(s) |
|---|---|---|---|---|
| $Y_1^{0,\mathrm{ff}}(\cdot,\cdot)$ | Fossil fuel | Open-source Data Inventory for Anthropogenic $CO_2$ 2016 (ODIAC2016) & Temporal Improvements for Modeling Emissions by Scaling (TIMES) | Monthly + hourly scaling | Oda and Maksyutov (2011); Nassar et al. (2013); Oda et al. (2018) |
| $Y_1^{0,\mathrm{bio}}(\cdot,\cdot)$ | Biospheric | CarbonTracker 2019, based on the Carnegie-Ames Stanford Approach (CASA) biogeo-chemical model | 3 hourly | Potter et al. (1993); Giglio et al. (2013); Jacobson et al. (2020) |
| $Y_1^{0,\mathrm{bf}}(\cdot,\cdot)$ | Biofuels | Yevich & Logan | Constant | Yevich and Logan (2003) |
| $Y_1^{0,\mathrm{fire}}(\cdot,\cdot)$ | Fires | Quick Fire Emissions Dataset, version 2 (QFED2) | Hourly | Darmenov and da Silva (2015) |
| $Y_1^{0,\mathrm{ocn}}(\cdot,\cdot)$ | Ocean | Scaled Takahashi | Monthly | Takahashi et al. (2002); Takahashi et al. (2009) |

## 4.2 Basis functions

We divided the globe into the $r_s = 22$ disjoint TransCom3 regions (Gurney et al., 2002), and time into the $r_t = 31$ months between (and including) September 2014 and March 2017, and then we constructed one flux basis function for each region/month pair. This yielded $r = r_s \times r_t = 682$ basis functions, each with non-zero support in a space-time volume spanning one month in time, and one TransCom3 region in space. Half of the TransCom3 regions are land regions, and half are ocean regions, so that half of our basis functions correspond to land areas, and half to ocean areas. We show the TransCom3 regions in Fig. C1, and their codes and labels in Table C1, both of which can be found in Appendix C. Some areas of the globe, depicted in white in Fig. C1, are assumed to have zero flux; all our basis functions are zero in these regions.

For $\phi_j(\cdot,\cdot)$ a basis function corresponding to a land area, we have

$$\phi_j(\mathbf{s},t) = \begin{cases} Y_1^{0,\mathrm{bio}}(\mathbf{s},t) + Y_1^{0,\mathrm{bf}}(\mathbf{s},t) + Y_1^{0,\mathrm{fire}}(\mathbf{s},t), & (\mathbf{s}',t)' \in D_j^\phi, \\ 0 & \text{otherwise,} \end{cases} \quad (14)$$





where $D_j^\phi$ is the space-time volume over which the $j$th basis function is defined to be non-zero. For $\phi_j(\cdot,\cdot)$ a basis function corresponding to an ocean area, we have

$$\phi_j(\mathbf{s},t) = \begin{cases} Y_1^{0,\mathrm{ocn}}(\mathbf{s},t), & (\mathbf{s}',t)' \in D_j^\phi, \\ 0 & \text{otherwise.} \end{cases} \tag{15}$$

Note that both Eq. (14) and Eq. (15) exclude $Y_1^{0,\mathrm{ff}}(\cdot,\cdot)$. This is done to meet the MIP requirement that fossil-fuel fluxes are
treated as fixed and known (which is common practice in flux inversion; e.g., see Basu et al., 2013). The influence of fossil fuels on the mole-fraction field is therefore present only as an invariant component of the prior expectation of the mole-fraction field.

Since the spatio-temporal patterns of the fluxes within a region–month space-time volume are fixed, our construction is quite restrictive. However, the space-time patterns are dictated by those in the inventories used to construct the basis functions, and
they can be assumed to be fairly realistic. This underlying assumption is ubiquitous in flux-inversion systems; for example, Jacobson et al. (2020) scale 3-hourly fluxes using weekly scale factors over 156 regions, while Basu et al. (2013) use monthly scale factors for 3-hourly fluxes over $6° \times 4°$ grid cells. Constraining the spatio-temporal pattern is inferentially advantageous because it helps address the ill-posed nature of flux inversion. It is also computationally advantageous because it reduces the number of unknowns for which inference is needed. The disadvantage is that the reliance on a priori structures increases the risk
of dimension-reduction error because, while our basis functions allow the posterior fluxes to vary at sub-TransCom3-region scales, variations that don't follow the prescribed pattern are necessarily ignored.

As described in Sect. 2.2, for $j = 1, \ldots, 682$, each flux basis function $\phi_j(\cdot,\cdot)$ has a corresponding mole-fraction basis function $\hat{\psi}_j(\cdot,\cdot,\cdot)$, which may be constructed by running the transport model, $\hat{\mathcal{H}}$, under the flux $Y_1^0(\cdot,\cdot) + \phi_j(\cdot,\cdot)$. Then the mole-fraction basis function $\hat{\psi}_j(\cdot,\cdot,\cdot)$ is recovered through linearity by simply subtracting $Y_2^0(\cdot,\cdot,\cdot)$ from the output mole-fraction
field. For illustration, Fig. 2 shows examples of the flux basis functions (monthly averaged) for January 2016 and the regions TransCom3 02 and TransCom3 06, and snapshots of the corresponding mole-fraction basis functions (daily and atmospheric-column averaged) obtained using the transport model described next in Sect. 4.3.

## 4.3 Transport model and initial condition

For our approximate transport model, $\hat{\mathcal{H}}$, we used the GEOS-Chem global 3-D chemical transport model, version 12.3.2 (Bey
et al., 2001; Yantosca, 2019), driven by the GEOS-FP meteorological fields from NASA's Goddard Earth Observing System (Rienecker et al., 2008). We use the offline GEOS-Chem $CO_2$ simulation (Nassar et al., 2010), with the native horizontal resolution of $0.25° \times 0.3125°$ and 72 vertical levels aggregated to $2° \times 2.5°$ and 47 vertical levels for computational efficiency. We use a transport time step of 10 minutes, and a flux time step of 20 minutes. All fluxes described in sections 4.1 and 4.2 were implemented in GEOS-Chem using the HEMCO emissions component (Keller et al., 2014). GEOS-Chem can be configured
to allow for a 3-D chemical source of $CO_2$ due to oxidation of other trace gases, but this was disabled for compatibility with the OCO-2 MIP.



**Figure 2.** Examples of flux basis functions that have support in the month of January 2016 in the regions TransCom3 02 and TransCom3 06, and the corresponding mole-fraction basis functions. The first row shows the values of the flux basis functions, $\phi_j(\cdot,\cdot)$, averaged over the whole month (these basis functions are zero outside January 2016). The next three rows show daily averages of the column-averaged $CO_2$ for the corresponding mole-fraction basis functions on three days: the start of the month-long period where the flux is non-zero, 01 January 2016; the middle of the period, 15 January 2016; and 15 days after the end of the period, 15 February 2016.





The approximate initial condition, $\hat{Y}_2(\cdot, \cdot, t_0)$, specifies the mole-fraction field at the beginning of the study period on 01 September 2014. For our initial mole-fraction field, we used a modified version of that generated by Bukosa et al. (2019). This mole-fraction field was constructed using a spin-up period, starting on 01 January 2005, and ending on 01 September 2014,

with transport driven by inventory fluxes and meteorology (that in some cases differ from those we use here; see Bukosa et al., 2019, for details). At the end of the spin-up period, the whole mole-fraction field on 01 September 2014 was scaled such that the value in the surface grid cell containing the South Pole was equal to the monthly-averaged mole-fraction measurement from surface flask measurements at the South Pole (Thoning et al., 2020) in September 2014. The South Pole was chosen as our calibration point due its physical isolation from strong sources and sinks.

**4.4    Data**

This study uses a subset of the data sources in the MIP (Crowell et al., 2019). These include retrievals of column-averaged $CO_2$ by NASA's OCO-2 satellite (Eldering et al., 2017), and retrievals of column-averaged $CO_2$ from TCCON (Wunch et al., 2011a). As in the MIP, we use OCO-2 data to estimate $CO_2$ fluxes, and TCCON data to validate the estimates.

**4.4.1    OCO-2**

The OCO-2 satellite was launched in 2014 with the goal of retrieving atmospheric $CO_2$ mole fractions. The on-board instrument measures radiances in three near-infrared spectral bands, which in turn are used to retrieve the $CO_2$ mole fraction on 20 vertical levels via a retrieval algorithm based on Bayesian optimal estimation (Rodgers, 2000; O'Dell et al., 2012). The retrieved levels are column-averaged, and then bias-corrected through comparison with TCCON retrievals (Wunch et al., 2011a). The OCO-2 team releases regular revisions of the retrieval dataset.

OCO-2 radiance measurements are taken in three distinct pointing modes: nadir mode, where the satellite aims at the point directly beneath it; glint mode, where the satellite points at the reflection of the sun on the surface; and target mode, where the satellite aims at a specific target, typically a ground station that also measures $CO_2$ mole fractions. Target observations are generally excluded from flux inversions and used only for instrument calibration. Over the ocean, nadir measurements have insufficient signal-to-noise ratio to provide useful retrievals, while over land, both nadir and glint retrievals are made. There

are therefore three retrieval modes to consider, land glint (LG), land nadir (LN), and ocean glint (OG). The error properties of retrievals over land differ significantly from those over ocean; in particular, the OG retrievals in the V7r dataset (the dataset used in the MIP) are not considered reliable, and were therefore excluded from the MIP (Crowell et al., 2019). We follow this protocol, and only do inversions using LG and LN data.

The MIP protocol dictated the use of a post-processed version of the V7r retrievals; this post-processing was done as follows.

First, an additional bias-correction term related to high-albedo measurements was applied to the XCO2 retrievals. The bias-corrected retrievals were then grouped and averaged into 1 s bins, and then they were further grouped and averaged into 10 s bins. The 10 s spans correspond to ground swathes of approximately 67 km in length. The standard deviation for each 10 s retrieval was computed as a function of the individual retrieval standard deviations, with an additional model–data mismatch term added to account for the expected differences arising from transport-model errors. For the MIP, the 10 s averages were





assumed to be independent but, following Sect. 2.4, we treat them as dependent. For more details on how the 10 s averages were computed, including how the standard errors were derived, see Crowell et al. (2019).

The OCO-2 retrieval algorithm produces estimates of $XCO_2$. In Eq. (7), we encapsulate the retrieval algorithm in the observation operator, $\hat{\mathcal{A}}_i$. Appendix B gives more details on this observation operator in the case of OCO-2 retrievals.

### 4.4.2 TCCON

TCCON is a network of ground-based sites measuring solar radiances in the near-infrared spectral band (Wunch et al., 2011a). Similar to the way OCO-2 retrievals are obtained, these measurements are converted to retrievals of column-averaged $CO_2$ (and other gases) using a retrieval algorithm. TCCON retrievals have been adjusted to agree with World Meteorological Organization (WMO) trace-gas measurement scales, and validated using aircraft data (Wunch et al., 2010). As both TCCON and remote sensing instruments retrieve column-averaged mole fractions, the TCCON data are an important validation resource (Wunch et al., 2011b). The MIP used TCCON column-averaged $CO_2$ retrievals from the GGG2014 release as validation data, including all retrievals available as of July 6, 2017. In the MIP, outliers and retrievals corresponding to soundings with high solar zenith angle were removed. The remaining retrievals were then averaged over 30-minute intervals; further details are given by Crowell et al. (2019). We note that the filtering procedure used in the MIP occasionally led to long periods of time for which data were considered missing. For consistency with the MIP, in this study we used the same retrievals and postprocessing methods; the stations used are listed in Table 2.

Like OCO-2, TCCON retrievals also have an associated observation operator $\hat{\mathcal{A}}_i$. This has a similar form to the operator for OCO-2, which is described in Appendix B. A detailed description of the TCCON observation operator is given by Wunch et al. (2011b).

### 4.5 Prior distributions over the parameters

The prior distributions for the parameters governing the scaling factors, $\boldsymbol{\alpha}$, are specified separately for the land and ocean TransCom3 regions. The land regions, which are observed directly by OCO-2 when in LG or LN mode, are assigned a non-informative prior, while the indirectly observed ocean regions (which also have relatively small fluxes over a given area) are assigned a relatively informative prior. Informative priors for ocean fluxes were deemed necessary following OSSEs performed by us, which revealed that it is often not possible to reliably constrain ocean fluxes from OCO-2 land data.

Specifically, for $j$ corresponding to a land region, we assigned a prior to $\kappa_j$ by letting $a_{\kappa,j} = b_{\kappa,j} = 1$ (resulting in a uniform prior over $[0,1]$), and for $\tau_{w,j}$ we let $\nu_{w,j} = 0.354$ and $\omega_{w,j} = (1 - \kappa_j^2)/0.0153$. This prior on $\tau_{w,j}$ implies that $1/\tau_{w,j}$, the marginal variance of the elements of the scalings in $\boldsymbol{\alpha}$ that correspond to land regions, has 5% and 95% percentiles of 0.01 and 10, respectively, which is reasonably uninformative. For $j$ corresponding to an ocean region, we apply an independent and identically distributed Gaussian prior with mean zero and standard deviation 0.5 to $\alpha_j$. This is achieved by fixing $\kappa_j = 0$ and $\tau_{w,j} = 4$.

As described in Sect. 4.4.1, the OCO-2 MIP 10 s averages come with prescribed uncertainties that include both measurement error and transport-model error. In our framework, the parameters governing these error processes are $\gamma_g$, $\rho_g$, and $\ell_{g,1}$. For the





**Table 2.** Names of the TCCON stations used in this study.

| Name | Reference |
| --- | --- |
| Ascension Island, Saint Helena | Feist et al. (2014) |
| Bialystok, Poland | Deutscher et al. (2015) |
| Bremen, Germany | Notholt et al. (2014) |
| Caltech, Pasadena, CA, USA | Wennberg et al. (2015) |
| Darwin, Australia | Griffith et al. (2014a) |
| Edwards (Armstrong/Dryden), CA, USA | Iraci et al. (2016) |
| Eureka, Canada | Strong et al. (2016) |
| Izaña, Spain | Blumenstock et al. (2014) |
| Karlsruhe, Germany | Hase et al. (2015) |
| Lamont, OK, USA | Wennberg et al. (2016) |
| Lauder, New Zealand | Sherlock et al. (2014) |
| Manaus, Brasil | Dubey et al. (2014) |
| Orléans, France | Warneke et al. (2014) |
| Park Falls, WI, USA | Wennberg et al. (2014) |
| Réunion Island, France | De Mazière et al. (2014) |
| Saga, Japan | Kawakami et al. (2014) |
| Sodankyla, Finland | Kivi et al. (2014) |
| Tsukuba, Japan | Morino et al. (2016) |
| Wollongong, Australia | Griffith et al. (2014b) |

prior distribution of $\gamma_g$, we let $\nu_{\gamma_g} = 1.627$ and $\omega_{\gamma_g} = 2.171$, which lead to 5% and 95% prior percentiles of 0.5 and 10, respectively, while we used a uniform prior distribution for $\rho_g$. For $\ell_{g,1}$, we let $\nu_{\ell_g,1} = 1$ and $\omega_{\ell_g,1} = 1$ min$^{-1}$. When doing
bias correction online, we used the prior on $\boldsymbol{\beta}$ described in Sect. 2.4.

### 4.6 Computation

Computations were performed in two stages. In the first stage, the 682 mole-fraction basis functions were precomputed in the manner described in Sect. 4.2. This is the most computationally demanding step, as each basis function requires the CTM to be run for several days of clock time, on average. Fortunately, since every basis function can be computed independently from all
others, computing them is an embarrassingly parallel problem. Furthermore, since the basis functions are shared between the different inversions in this section, they only need to be computed once. Computation of the basis functions took seven days in total using the Gadi supercomputer at the Australian National Computational Infrastructure.





The inversions were performed in the second stage. As mentioned in Sect. 2.5, the posterior distribution for each inversion was estimated using an MCMC sampling scheme, with details given in Appendix A. The sampling schemes in all cases were
run for 11,000 iterations, and the first 1,000 iterations were discarded as burn-in. Convergence of the MCMC chain was assessed by inspection of all the trace plots. The scheme was implemented in the R programming language (R Core Team, 2020), with intensive linear algebraic computations offloaded for performance to a graphics processing unit (GPU) using Tensorflow (Abadi et al., 2016). The total running time of the sampler depended on which model assumptions were used; in particular, whether uncorrelated (15 minutes) or correlated errors (two hours) were modelled. All inversions were performed on a machine with
an 8-core Intel i9-9900K CPU running at 3.60 GHz and an NVIDIA RTX 2080 GPU.

## 5 Results

In this section we evaluate WOMBAT, first in an OSSE, where the true fluxes are assumed known and data are simulated from these true fluxes, then on actual satellite data via the MIP protocol of Crowell et al. (2019). Using an OSSE, described in Sect. 5.1, serves two purposes: first, to show that WOMBAT can indeed recover the true fluxes in a controlled environment
where the "working model" is the "true model"; and second, to illustrate the importance of modelling measurement-error biases and correlated errors when these are present in the true model from which the data are simulated. Then, in Sect. 5.2, we show that WOMBAT gives similar flux estimates to those obtained by different MIP participants, and that it performs well relative to the MIP participants in reproducing out-of-sample TCCON validation data. In Sect. 5.3 we show that WOMBAT is able to estimate bias coefficients online, if needed.

**5.1 Observing system simulation experiment (OSSE)**

In this section we illustrate the use of WOMBAT in an OSSE, where we randomly draw flux scaling factors $\boldsymbol{\alpha}^s$ from a Gaussian distribution with mean $\mathbf{0}$ and covariance matrix $0.09\mathbf{I}$, and assume that these are the true flux scaling factors. The (simulated) true flux $Y_1^s(\cdot,\cdot)$ is given by

$$Y_1^s(\cdot,\cdot) = Y_1^0(\cdot,\cdot) + \sum_{j=1}^{r} \phi_j(\cdot,\cdot)\alpha_j^s, \tag{16}$$

where $\alpha_j^s$ is the $j$th element of $\boldsymbol{\alpha}^s$. The (simulated) true mole-fraction field, $Y_2^s(\cdot,\cdot,\cdot)$, is then given by

$$Y_2^s(\cdot,\cdot,\cdot) = Y_2^0(\cdot,\cdot,\cdot) + \sum_{j=1}^{r} \hat{\psi}_j(\cdot,\cdot,\cdot)\alpha_j^s. \tag{17}$$

Finally, we simulate measurements from the mole-fraction data model in Eq. (9) at the same times and locations as the OCO-2 10 s average retrievals for the LN and LG modes, by passing $Y_2^s(\cdot,\cdot,\cdot)$ through the corresponding OCO-2 observation operators (see Sect. 4.4.1).

When simulating data via Eq. (9), we assume that both $\mathbf{b}_g$ and $\boldsymbol{\xi}_g$ are present. For the bias term $\mathbf{b}_g$, we assume that the OCO-2 retrieval biases are a linear combination of covariates that are associated with bias in the retrieval process:





– "dp": The prior–retrieval surface pressure differential;

– "logDWS": The logarithm of the total retrieved optical depth associated with the aerosol types dust, water cloud, and sea salt; and

– "co2_grad_del": The difference between the retrieved $CO_2$ mole fractions at the surface and retrieval vertical level 13 (corresponding to the height with air pressure equal to 63.2% of the surface pressure, around 520–650 hPa for most retrievals).

The "official" V7r bias-correction parameters (regression coefficients) for the original Level 2 (L2) data release were obtained through offline comparison of the raw L2 product with TCCON retrievals, and they are the same for both LG and LN observa-

tions. They are equal to 0.3, 0.028, and 0.6, for the three variables, respectively. We construct our (simulated) true biases based on these coefficients.

As discussed in Sect. 2.4, we assume that the prescribed variance of each retrieval needs to be inflated, and the inflated variance is the sum of the variance from both the correlated ($\boldsymbol{\xi}_g$) and uncorrelated ($\boldsymbol{\epsilon}_g$) error components. In our OSSE, we assume that the inflation factor of the prescribed variances, $\mathbf{v}_g^{\mathrm{ps}}$, is $\gamma_g = 1.25$, and that the proportion of this variance allocated

to the correlated error process is $\rho_g = 0.8$. We induce the correlations using the exponential covariance function described in Sect. 3 with the single length scale of the correlated component set to $\ell_{g,1} = 1$ minute for all $g = 1, \ldots, n_g$. We specify this correlation structure to be the same for both LG and LN data.

We ran four different setups in WOMBAT, where bias is assumed or not assumed to be present, and where errors are assumed or not assumed to be correlated. The known true flux, generated as described above, is the same between the cases,

and we evaluate the ability of WOMBAT to recover the truth under each of the setups. Table 3 gives the root-mean-squared error (RMSE) and continuous ranked probability score (CRPS, Gneiting and Raftery, 2007) when estimating monthly- and regionally-aggregated fluxes using these four setups. The regions on which these evaluations are based are the same TransCom3 regions that were used to construct the flux basis functions (see Sect. 4.2), and the quantities in the table are averages across all combinations of the 31 months and 22 regions. The true data-generating process involves both bias and correlated error. There-

fore, as one would would expect, Table 3 shows that the WOMBAT setup that takes into account both of these features performs the best in terms of both RMSE and CRPS, while the setup that assumes that neither feature is present performs the worst. For the two partially misspecified setups, the bias-corrected/uncorrelated setup outperforms the not-bias-corrected/correlated setup for LG data, while the opposite is true for LN data. Notably, despite the presence of systematic biases in the simulated data, the WOMBAT setup that assumes no bias, but which takes into account correlated errors, performs overwhelmingly better than

the fully misspecified model that assumes no bias and uncorrelated errors.

Figure 3 shows the (simulated) true (black), prior mean (blue), and posterior distributions (red, purple, orange, and green), for the total flux in the tropics (latitude $23.5°$ S–$23.5°$ N) in 2015 and 2016, from both LN and LG, split by the southern and the northern components (latitudes $23.5°$ S–$0°$ and $0°$–$23.5°$ N, respectively). The four posterior distributions depicted in each panel correspond to the four different WOMBAT setups. The interior of the ellipses represent the 95% credible regions. The

grey dotted lines along the diagonal correspond to combinations of the southern and northern fluxes that yield the true total




**Table 3.** Root-mean-squared error (RMSE) and continuous ranked probability score (CRPS) when estimating monthly regional fluxes using LG and LN data in the OSSE of Sect. 5.1. The lower the error or the score, the better the performance. Four setups in WOMBAT are evaluated and the regions and time periods over which these summaries (averages) are obtained are the same as those used for constructing flux basis functions (see Sect. 4.2).

| | RMSE [PgC mo$^{-1}$] | | CRPS | |
| Configuration | LG | LN | LG | LN |
|---|---|---|---|---|
| Bias correction/correlated errors | 0.023 | 0.021 | 0.010 | 0.009 |
| Bias correction/uncorrelated errors | 0.038 | 0.038 | 0.015 | 0.016 |
| No bias correction/correlated errors | 0.045 | 0.026 | 0.016 | 0.011 |
| No bias correction/uncorrelated errors | 0.092 | 0.063 | 0.034 | 0.026 |
| Prior | 0.036 | 0.036 | 0.018 | 0.019 |

flux in the tropics; if the dotted line is not within the ellipse for an inversion, the total flux is misestimated. All fluxes shown in Fig. 3 are exclusive of fossil fuels which, recall, are held fixed in the inversions. Figure C2 in Appendix C shows the results on a global scale (land vs ocean), while Fig. C3 in Appendix C shows the results on a regional scale (TransCom3 region 04 [South American Temperate] vs TransCom region 06 [Southern Africa]) .

Collectively, the performances of the different models, as shown in Figs. 3, C2, and C3, align with the conclusions based on the CRPS and RMSE statistics. In all the cases shown, the 95% credible regions for the WOMBAT configuration with both bias correction and correlated error (red) contain the true value, while those for the configuration with neither feature (green) do so rarely. The orange credible regions, for the bias-corrected/uncorrelated variant, are always smaller than the red credible regions, indicating that the bias-corrected/uncorrelated variant is overconfident. In contrast, the purple credible regions, for the

not-bias-corrected/correlated variant, are always larger, which may suggest that the correlated errors are partially compensating for the lack of bias correction in this variant.

     In summary, this OSSE shows that WOMBAT can recover the true flux when the assumed model is the true model. But, more importantly, the OSSE also demonstrates the importance of modelling both bias and correlated errors in these flux inversions. If the bias parameters are omitted, fluxes can be estimated incorrectly, although this may be partially mitigated by modelling

correlated errors. If uncorrelated errors are assumed, estimation performance suffers, and flux estimates will likely be reported with too small an uncertainty, even if the prescribed variances are allowed to be inflated when making inference. In a real-data setting, any factors thought to introduce systematic biases should be taken into account, but this OSSE also suggests that the use of correlated errors may provide some insurance against any remaining, unmitigated, spatio-temporal biases.

### 5.2   OCO-2 satellite data

In this section we present results from WOMBAT applied to OCO-2 satellite data under the MIP protocol (Crowell et al., 2019). The protocol mandates the use of OCO-2 retrievals with the TCCON-based offline bias correction. While WOMBAT



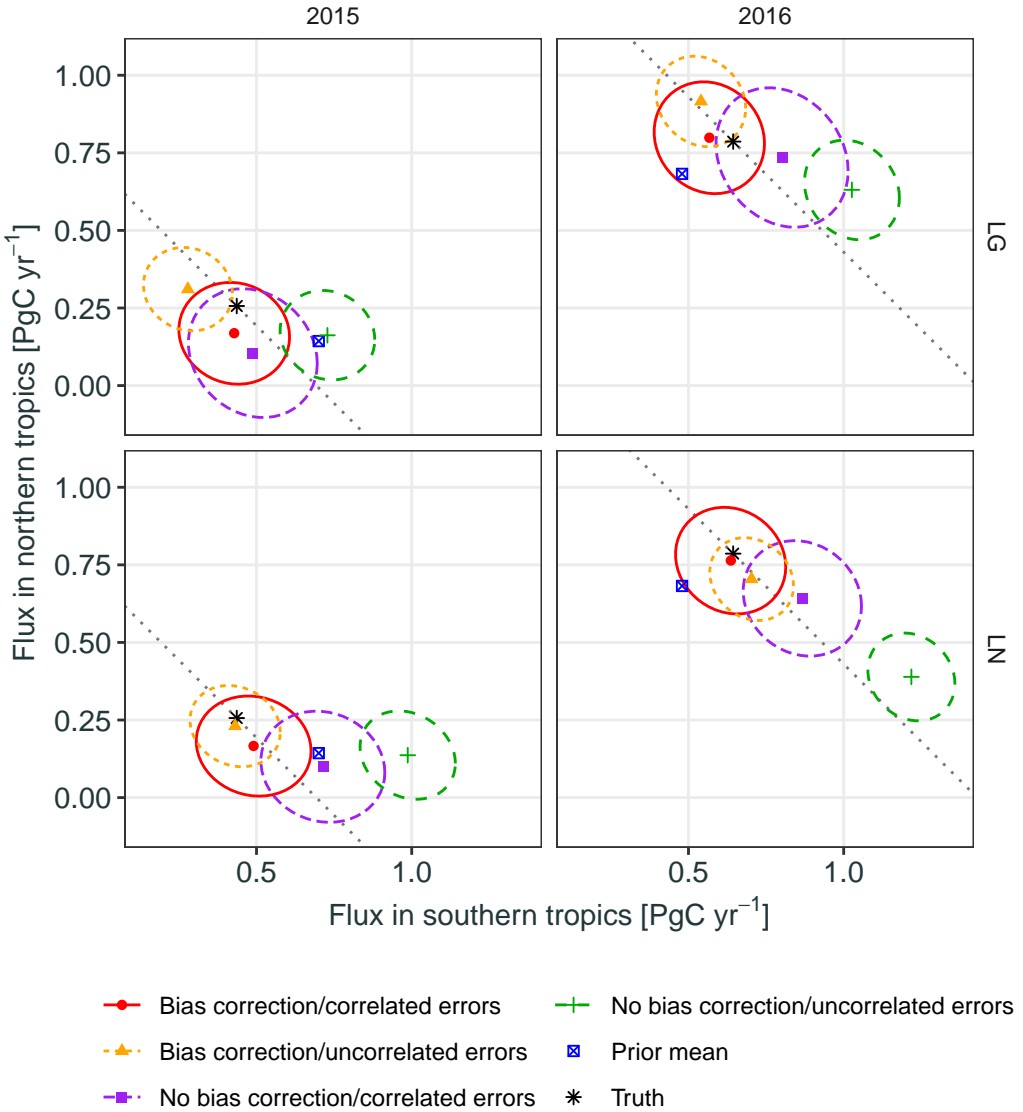

**Figure 3.** True, prior, and posterior, estimates of total flux in the northern tropics ($0°$–$23.5°$ N) versus the total flux in the southern tropics ($23.5°$ S–$0°$) for the OSSE in Sect. 5.1. The columns correspond to the years 2015 and 2016, while the rows show which observation groups were used, either OCO-2 land glint (LG) or land nadir (LN), noting that the prior and true flux are the same across rows. Points show the posterior mean fluxes for each model configuration, as well as the prior mean in blue and the truth in black. The ellipses contain 95% of the posterior probability for the true tropical fluxes in the southern and northern hemispheres. The grey dotted lines along the diagonal correspond to combinations of southern and northern tropical fluxes that yield the true total flux in the tropics. All fluxes are exclusive of fossil fuels, which are held fixed in the inversion.





is capable of online bias correction (see Sect. 5.3), in this section we follow the MIP protocol and set the bias parameters in WOMBAT equal to zero.

### 5.2.1 Flux comparison on a region–time basis with the MIP

In the OCO-2 MIP, nine participants submitted fluxes based on inversions satisfying the MIP protocol. Each participant reported to the MIP four sets of fluxes: their prior mean fluxes, and their fluxes from three inversions based respectively on point referenced data, OCO-2 LG data, and OCO-2 LN data. Crowell et al. (2019) considered the different participants' fluxes as an ensemble, reporting the ensemble mean, median, and standard deviation across a variety of temporal and spatial scales. Under the same protocol and using OCO-2 LG and OCO-2 LN data, we compare WOMBAT's posterior distribution over the fluxes

to the corresponding results from the MIP ensemble.

Through its MCMC Bayesian computations, WOMBAT's inversions generate samples from the posterior distribution of all unknown quantities in the model, including the parameters discussed in Sect. 2.4. Section 5.2.3 discusses the inferred parameters in detail for inversions corresponding to different satellite modes. Of note are the posterior distributions of the parameters $\{\rho_g\}$, which are centred on values around 0.84. This is a strong indication that the majority of the combined model–

data discrepancy/measurement-error process should indeed be attributed to the correlated component, $\boldsymbol{\xi}_g$, given in Eq. (9). Samples from the MCMC scheme enable estimation of functionals of the posterior distribution, including posterior means and quantiles, of the flux process $Y_1(\cdot, \cdot)$. While some individual MIP participants are able to produce probabilistic uncertainty estimates for fluxes, these were not reported as part of the MIP; instead, the empirical distribution from the ensemble of fluxes was used by the MIP for uncertainty quantification. Since it is difficult to make a quantitative comparison between WOMBAT's

posterior-based uncertainties and the ensemble uncertainties in the MIP, we opt here for a visual comparison of the posterior means and standard deviations over the fluxes compared to the ensemble minimum, mean, and maximum. This comparison is done at both annual and monthly temporal scales, and across spatial scales encompassing the whole globe, global land, global ocean, and zonal bands.

**Global totals:** Figure 4 presents annual and monthly non-fossil-fuel fluxes for the globe, land regions, and ocean regions.

Fluxes are shown for the MIP, split into prior, LG inversions, and LN inversions and, for WOMBAT, they are split into prior, posterior using LG data, and posterior using LN data. Thick lines show the ensemble means for the MIP, and the (prior or posterior) means for WOMBAT. Shaded areas and thin lines for the MIP denote the values between the ensemble minimum and maximum, while for WOMBAT they denote values in the 95% fully Bayesian credible intervals.

The global annual non-fossil-fuel fluxes estimated by posterior means from WOMBAT are very similar for both LG and

LN modes, with an overall posterior mean net sink in 2015 of 3.40/3.14 PgC yr$^{-1}$ for LG/LN, respectively, and in 2016 of 4.14/4.27 PgC yr$^{-1}$ for LG/LN, respectively. For 2015, these sinks are very similar to the MIP ensemble means (3.57/3.21 PgC yr$^{-1}$ for LG/LN, respectively). However, for 2016, WOMBAT returns a deeper posterior-mean sink than the MIP ensemble means (3.82/3.78 PgC yr$^{-1}$ for LG/LN, respectively), and its 95% credible intervals do not contain within them the ensemble means. At a monthly scale, WOMBAT reproduces a key feature of the MIP fluxes, wherein the seasonal cycle in the fluxes,

driven by the Northern Hemisphere growing season, begins and ends earlier than it does in the prior for both 2015 and 2016. In



**Figure 4.** Annual (left column) and monthly (right column) fluxes for the globe (first row), land (second row), and ocean (third row). Flux estimates from both the MIP and WOMBAT are shown, split into the prior, LG inversions, and LN inversions. Thick lines represent the ensemble means for the MIP and the (prior or posterior) means for WOMBAT. Shaded areas and thin lines for the MIP represent values between the ensemble minimum and maximum, while for WOMBAT they represent values in the 95% credible intervals (cred. int.). Fossil-fuel fluxes are excluded from all figures. Note that each row of plots has a different vertical scale.





agreement with the MIP, WOMBAT results indicate that the deepest sink in the cycle is larger than that in the prior. However, the sink estimated by WOMBAT is around 0.4 PgC mo$^{-1}$ smaller than the MIP ensemble mean for both LG and LN and for both 2015 and 2016.

**Global land and ocean:** For global land fluxes, shown in the second row of Fig. 4, WOMBAT's results agree with those
from the MIP for both LG and LN in that a source larger than that in the prior flux is estimated for October 2015. However, while the source persists into November in the MIP ensemble mean, the WOMBAT posterior mean does not have the same persistence. For global ocean fluxes, shown in the third row of Fig. 4, the MIP LG-estimated and LN-estimated fluxes differ little from the prior fluxes, and we observe the same for global oceans in the WOMBAT estimates for both LG and LN modes and in most months. The exceptions are September and October in both years, where WOMBAT estimates a shallower sink,
and even zero flux with the LN data. These features are not obvious in the MIP ensemble means, but they do appear reasonable within the MIP ensemble spread.

**Zonal bands:** Figure C4 in Appendix C shows fluxes for zonal bands covering the northern extratropics (23.5° N–90° N), northern tropics (0°–23.5° N), southern tropics (23.5° S–0°), and southern extratropics (90° S–23.5° S). For the MIP ensemble, Crowell et al. (2019) noted that inversions using LG data led to a smaller net annual sink (averaged between 2015 and 2016) in
the northern extratropics than those using LN data. WOMBAT also finds this feature, with a 95% credible interval of the LG-minus-LN difference spanning 0.14–0.6 PgC yr$^{-1}$. This is substantially smaller than the difference between the MIP ensemble means for these modes, which is 0.7 PgC yr$^{-1}$. Fluxes in the southern extratropics, shown in the fourth row of Fig. C4, are dominated by ocean fluxes for which, as noted above, LG and LN data provide little information.

One of the most prominent features in the MIP inversion results is a seasonal cycle in the tropics that is larger than that in
both the prior mean and the in situ inversions (Crowell et al., 2019). From the second and third rows of Fig. C4, which depict tropical-zone fluxes, it can be seen that WOMBAT does not reproduce this for both LG and LN inversions. In the northern tropics, the WOMBAT posterior means are similar to the prior means, and the credible intervals in the annual fluxes reflect high confidence. However, results from WOMBAT do corroborate those of the MIP ensemble, in that non-fossil-fuel fluxes in the northern tropics were a net source of CO$_2$ in 2016.

**5.2.2 TCCON comparison**

To evaluate the estimated fluxes in the OCO-2 MIP, each participant was asked to use the 30-minute-average TCCON retrievals of column-averaged CO$_2$ (see Sect. 4.4.2) as validation data, and compare them to the column-averaged CO$_2$ predicted values obtained by applying the process model to the estimated fluxes with the same CTM used for the inversion. Recall that, when performing the inversions, only OCO-2 data were used, and the TCCON data were treated as unobserved and set aside for
validation. For WOMBAT, we repeated this validation exercise by examining the prior and posterior distributions of $\mathbf{Z}_{2,g}$, where each $g$ corresponds to a different TCCON site. For each group $g$, we set $\mathbf{b}_g = \mathbf{0}$, since we assume that the TCCON retrievals provided are free of bias. On the other hand, we assumed that $\boldsymbol{\xi}_g + \boldsymbol{\epsilon}_g$ has variance that is group-specific and that these errors are fully correlated. While this assumption is conservative, it is also reasonable, since the CTM does induce errors that are highly correlated in time at a common spatial location, as it averages all variables on a rather coarse grid when





**Table 4.** Mean-squared error (in ppm$^2$) averaged across TCCON stations, for each MIP participant and for WOMBAT's prior and posterior mean predicted values.

| | TM5-4DVAR | CT-NRT | OU | CAMS | Baker-mean | SCHUH | UT | CMS-Flux | UoE | WOMBAT Post. | WOMBAT Prior |
|---|---|---|---|---|---|---|---|---|---|---|---|
| LG | 1.33 | 1.63 | 1.31 | 1.31 | 1.88 | 1.41 | 1.85 | 1.78 | 1.71 | 1.19 | 3.56 |
| LN | 1.36 | 2.09 | 1.74 | 1.24 | 2.12 | 1.58 | 1.71 | 2.72 | 1.37 | 1.57 | 3.56 |

simulating atmospheric transport. We estimate the variance of these correlated errors in a group $g$ as the average of the reported variances of each TCCON retrieval within the $g$-th group.

In Fig. C5 in Appendix C, we compare the time series of the TCCON retrievals with the predictions from WOMBAT under the prior-mean flux, the posterior distribution of flux from LG data, and the posterior distribution of flux from LN data. Several things are of note from this figure: First, the posterior-mean estimates are a better match to the TCCON retrievals than the prior-

mean estimates, which is evidence that OCO-2 data do allow for improved flux estimates to be obtained. Second, discrepancies between TCCON and predicted retrievals persist for a long time, lending credence to our assumption that errors are highly temporally correlated. Third, the 95% prediction intervals are appropriate, and largely contain the TCCON retrievals.

In Fig. 5, we reproduce an augmented version of Fig. 8 of Crowell et al. (2019), which depicts the mean and standard deviation of the differences between the TCCON retrievals and the predicted retrieval by TCCON site, MIP participant, and

observation mode (LG and LN) alongside the results from WOMBAT. The improvement of the WOMBAT posterior prediction over the prior prediction is evident in the mean of the differences, and the posterior error means and standard deviations of WOMBAT are in line with those of the MIP participants. WOMBAT's predictive distributions from LG-inferred fluxes can be seen to be better than those of the MIP participants, even by straightforward visual inspection. In Table 4 we compute mean-squared error, by participant and observation mode, averaged over the 19 TCCON stations used in the MIP. WOMBAT

outperforms all other participants when using this metric with LG data, and it is fourth-best when using this metric with LN data. While these results are not conclusive on the validity of the WOMBAT fluxes globally, they are encouraging, especially in light of the fact that our flux process has a relatively low-dimensional representation.

### 5.2.3 The inferred parameters

One of the key features of WOMBAT is the use of a parameter prior distribution in the hierarchical Bayesian model, which ap-

plies to both the parameters governing the flux scaling factors and to the parameters governing the model–data discrepancy and measurement-error processes. Figure 6 shows the estimated posterior means and 95% credible intervals for the autoregressive parameters $\boldsymbol{\kappa}$ (top) and the innovation precisions $\boldsymbol{\tau}_w$ (bottom), for the 11 land regions, and for inversions using LG and LN data. The inferred parameters are relatively consistent across the LG and LN modes, with the exception of TransCom3 region

**Figure 5.** Mean (top row) and standard deviation (bottom row) of the errors across TCCON stations for each MIP participant (refer to Crowell et al., 2019, for details) as well as WOMBAT's prior and posterior mean predicted values. The error statistics for inversions using LG data are shown in the left column, while those for LN data are shown in the right column. This figure reproduces and extends Fig. 8 of Crowell et al. (2019) with similar (but not identical) colour gradients.

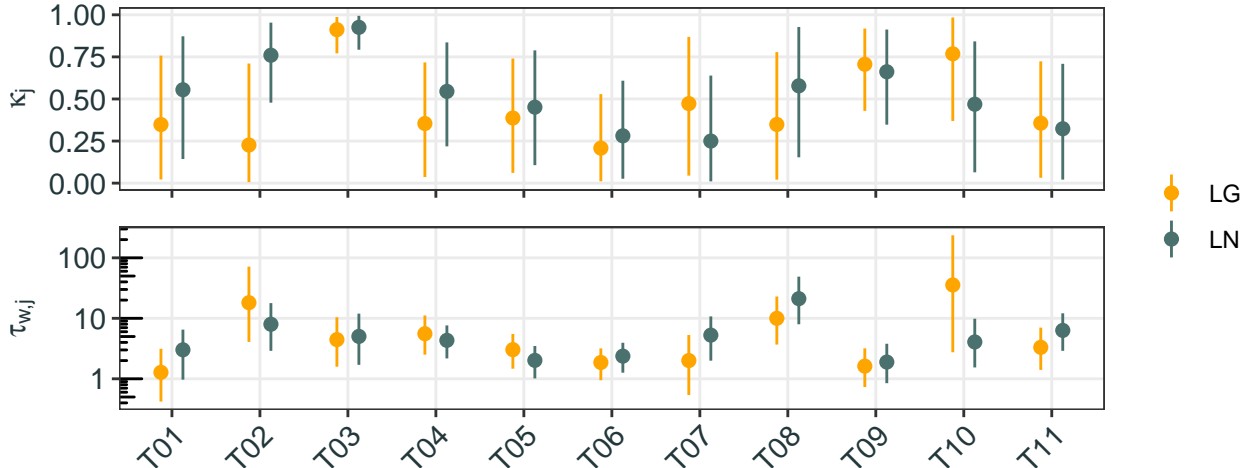

**Figure 6.** Posterior means and 95% credible intervals for $\kappa$ (top) and $\tau_w$ (bottom, shown using a log scale), for the 11 land regions: TransCom3 region 01 (T01) to TransCom3 region 11 (T11). Estimates are shown for inversions using LG data (yellow) and LN data (dark green).

02 (North American Temperate). Most regions have a posterior mean for $\kappa_j$ that is approximately between 0.25 and 0.75, re-
flective of moderate autocorrelation in the scaling factors. The exception is TransCom3 region 03, for which the scaling factors
are estimated to be highly autocorrelated a priori. The innovation precisions, $\tau_w$, have posterior means that lie approximately
between 1 and 10, for most regions.

The parameters governing the model–data discrepancy and measurement-error processes are $\rho_g$, $\ell_{g,1}$, and $\gamma_g$, for $g =
1, \ldots, n_g$. Table 5 gives the posterior means, 2.5% quantiles, and 97.5% quantiles, for these parameters. Recall that the LG
and LN parameters are derived from different inversions; they are not two groups in the same inversion. Nonetheless, the in-
ferred parameters are similar between the inversions, which is reassuring. The values for $\gamma_g$ are indicative of a 21% variance
inflation needed for both instrument modes. The length scales, $\ell_g$, are 1.2 minutes for the LG data and 1.1 minutes for the LN
data, which corresponds to around 700–800 km on the ground. Finally, the estimated values of $\rho_g$ are around 0.835, indicat-
ing that the majority of the combined model–data discrepancy/measurement-error process should indeed be attributed to the
correlated component, $\xi_g$, given in Eq. (9).

### 5.3 Online bias correction

The OSSE-based sensitivity study in Sect. 5.1 demonstrated that WOMBAT is able to perform *online* bias correction with
simulated data, where biases are estimated while doing flux inversion. This is different to the typical *offline* approach to bias
correction, where retrieval biases are determined in a separate study (e.g., Wunch et al., 2011b). To comply with the MIP





**Table 5.** Posterior means, 2.5% quantiles, and 97.5% quantiles, for the parameters $\ell_{g,1}$, $\gamma_g$, and $\rho_g$, for the inversions using LG and LN retrievals. Recall that the parameters associated with LG and LN are derived from different inversions, and not from using the two retrieval groups in the same inversion.

| | LG | | | LN | | |
|---|---|---|---|---|---|---|
| Variable | Mean | 2.5% | 97.5% | Mean | 2.5% | 97.5% |
| $\ell_{g,1}$ | 1.221 | 1.183 | 1.261 | 1.106 | 1.072 | 1.137 |
| $\gamma_g$ | 1.207 | 1.187 | 1.226 | 1.209 | 1.190 | 1.226 |
| $\rho_g$ | 0.834 | 0.829 | 0.838 | 0.835 | 0.831 | 0.840 |

protocol, the online bias-correction functionality of WOMBAT was disabled in the study of Sect. 5.2, and the TCCON-based offline bias-corrected OCO-2 retrievals from the MIP were used. In order to investigate the prospect of online bias correction with real data, we repeat the inversions with online bias correction enabled, using OCO-2 10 s average retrievals both with and without the TCCON-based offline corrections.

In Fig. 7, we show the posterior densities of the WOMBAT-estimated bias-correction coefficients when using the retrievals
without the offline correction. The posterior densities shown there are for inversions based on LG and LN retrievals, while the TCCON-based offline bias-correction coefficients are given by the blue vertical lines. The WOMBAT-estimated coefficients have the same sign and similar magnitudes to the offline corrections, suggesting that WOMBAT is picking up similar bias patterns while doing flux inversion. However, with the exception of the "dp" coefficient under the LN inversion, the offline values are outside the plausible ranges estimated by WOMBAT. For "co2_grad_del", WOMBAT favours a smaller correction
for both LG and LN, while for "logDWS" a larger correction is favoured. For "dp", WOMBAT favours a smaller correction under the LG inversion.

We repeated the online bias-correction procedure using retrievals retaining the TCCON-based offline bias correction. In this setting, if the retrievals are unbiased, bias coefficients equal to zero should be inferred. The posterior densities of the estimated coefficients under this configuration are shown in the second row of Fig. 7. As expected, the magnitudes of the online-estimated
coefficients are close to zero, although the credible intervals do not always include zero. Naïvely, one might expect that the coefficients would be approximately equal to the difference between the TCCON-based offline coefficients and the coefficients estimated by WOMBAT when using uncorrected data. For "dp" and "co2_grad_del", the estimated coefficients indeed have the expected sign, and the expected orders of magnitude. The inferred "logDWS" coefficients are surprising, however, with an opposite sign to what was expected for the LG inversion, and with smaller magnitudes for both the LG and LN inversions. This
unexpected result is a reflection of the complex interplay, and nonlinear relationships, between the parameters and processes in a regularised flux-inversion model.

Overall, the online-estimated bias correction is practically, if not statistically, similar to the TCCON-based offline correction. One possible explanation for the difference between the WOMBAT and the TCCON-based estimates is that different data are used, because WOMBAT does not use TCCON data for the correction. Moreover, it is likely that the true bias coefficients are



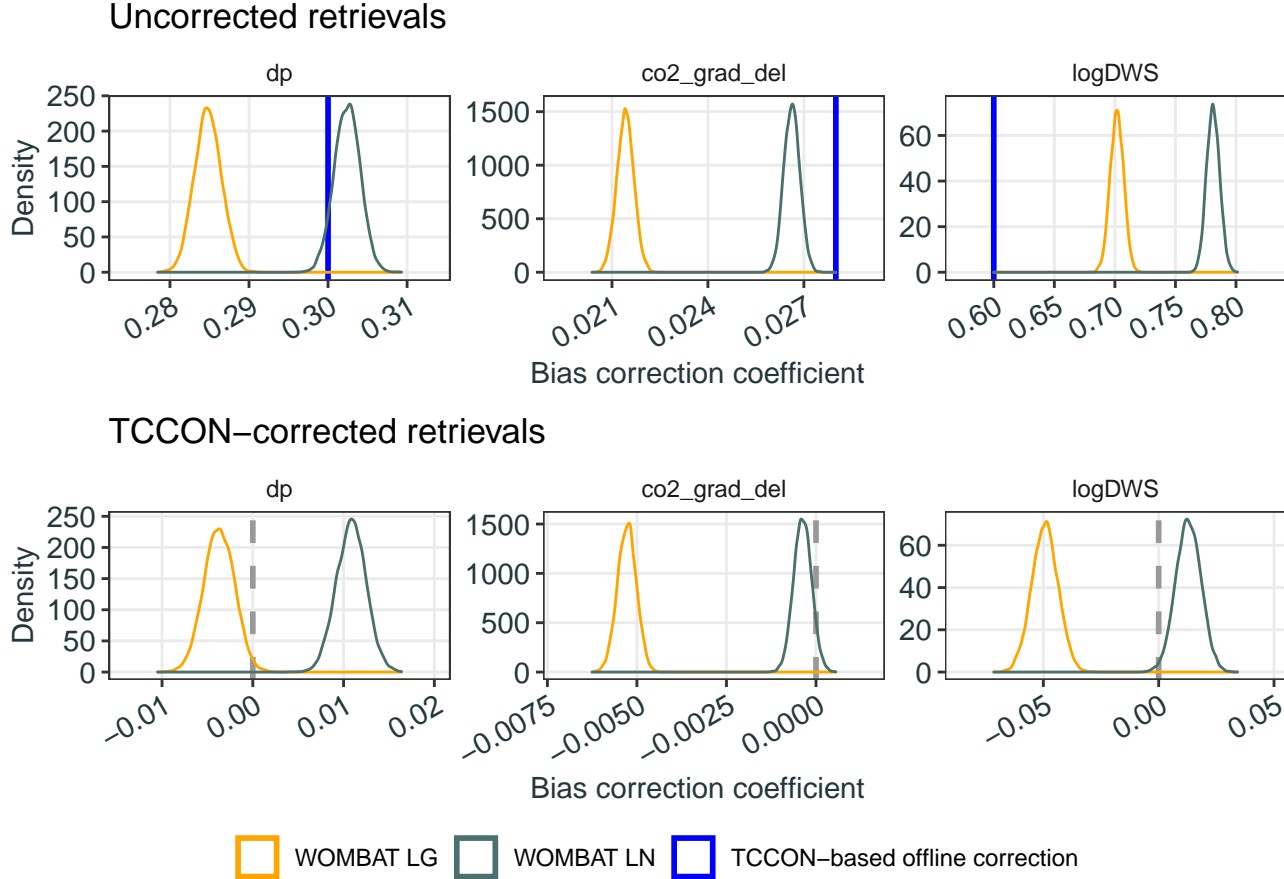

**Figure 7.** Posterior densities of bias-correction coefficients from inversions using OCO-2 retrievals without bias corrections applied (top row), and those from inversions using retrievals corrected using the TCCON-based offline bias coefficients (bottom row). Densities are shown for inversions based on LG and LN retrievals in yellow and dark green, respectively. The TCCON-based offline bias correction coefficients are shown as blue vertical lines for the panels in the top row. The grey vertical dashed lines in the bottom row mark zero, the value the coefficients would take if the data were unbiased.

spatio-temporally varying; if this is indeed the case, the estimated biases would be affected by the spatio-temporal locations of
the retrievals used to estimate them. Another reason may be that, for simplicity, we have used only a few of the most important
variables that are used for offline bias correction; a consideration of all the variables may lead to slightly different results.
Despite this, our results show that online bias correction is possible and that further research into it as an attractive by-product
of flux inversion is warranted.

Figure 8 gives annual and monthly fluxes estimated from the inversions using the online bias correction applied to the
uncorrected retrievals. For comparison, the equivalent fluxes from Sect. 5.2.1, for the offline-corrected data and with the online

**Figure 8.** Annual and monthly fluxes for WOMBAT inversions using: TCCON-based offline bias-corrected retrievals (solid lines with yellow for LG and dark green for LN; the fluxes are the same as in Fig. 4), and with online bias correction applied to the uncorrected retrievals (dashed lines). The prior-mean fluxes are shown in blue. Solid lines depict prior/posterior means, and shaded areas denote the 95% credible intervals (cred. int. ). The top row depicts global fluxes, the middle row global land fluxes, and the bottom row global ocean fluxes.

bias correction disabled, are also reported. The annual and monthly fluxes are similar between the offline-corrected and online-corrected inversions, with substantial overlap in the marginal posterior distributions for all time periods. This result gives further evidence that online bias correction is a viable alternative to offline correction (here based on TCCON data) when

doing flux inversion.





# 6 Conclusion

The WOllongong Methodology for Bayesian Assimilation of Trace-gases (WOMBAT) extends the standard synthesis flux-inversion framework, which does not put prior distributions on all unknowns, to a framework based on a fully Bayesian hierarchical statistical model. It incorporates physically motivated flux basis functions and follows the standard Bayesian synthesis framework by using a CTM to compute the corresponding mole-fraction basis functions offline. The scaling factors for the basis functions are inferred from mole-fraction satellite data. WOMBAT incorporates a correlated-error term, estimates measurement biases and measurement-error scaling factors online, estimates variances and length scales of flux scaling factors, and uses an MCMC scheme that allows uncertainty quantification through posterior distributions on all unknowns in the model. We have illustrated the importance of modelling correlation and bias within a flux-inversion system, and we have shown that WOMBAT produces global and regional flux estimates from OCO-2 data that are comparable to those from the MIP participants in Crowell et al. (2019). In particular, WOMBAT outperformed the other flux models in reproducing TCCON data when using the OCO-2 LG retrievals to obtain the fluxes, and it was competitive when using fluxes obtained from the OCO-2 LN retrievals. When the fossil-fuel fluxes are included, we estimate a global carbon source of $6.11 \pm 0.09$ PgC yr$^{-1}$ using the LG data, and $6.17 \pm 0.07$ PgC yr$^{-1}$ using the LN data. These estimates corroborate those of the MIP within uncertainty.

This paper presents the general, underlying Bayesian hierarchical framework for WOMBAT v1.0, which will serve as a baseline for our flux inversions based on current and future versions of OCO-2 data. There are several potential extensions being considered; here we discuss three of the most pertinent ones. First, WOMBAT, like most other flux-inversion systems, currently operates using a single CTM. This is problematic from a statistical modelling point of view, as it does not allow one to attribute the correlated error either to the measurement-error process or to the mole-fraction process. If more than one CTM is used, in principle one could statistically attribute at least part of the error due to transport. This will not necessarily solve the problem though, since CTMs tend to share common features that induce similar correlations (e.g., due to unresolved sub-grid variation). A possible way forward is to take results from offline OSSEs to estimate and fix the parameters characterising the CTM error, and then to attribute any residual observed correlation to the retrieval process.

Second, WOMBAT extends a traditional state-space approach to flux inversion, which competes with adjoint-based approaches that allow for a much higher flux dimensionality. Moving forward, WOMBAT will seek to introduce higher dimensionality by using flux basis function that are at a much finer scale than the TransCom3-by-month spatio-temporal basis functions that we have used here. These will help reduce any model error due to dimensionality reduction.

Finally, WOMBAT currently only considers along-track correlations when modelling the correlation-error term, $\boldsymbol{\xi}_g$. However, the general framework we have proposed, based on a sparse-precision-matrix approximation, can be extended to model full space-time correlations. Dedicated investigations will be required to assess the feasibility of the implementation and the impact it will have on flux estimates.





*Code and data availability.* The code associated with this article is available at https://doi.org/10.5281/zenodo.4886771 (Bertolacci et al., 2021). This code repository includes scripts for reproducing the entire analysis in this paper, which can be applied to a variety of inverse modelling problems. Re-running the entire analysis is computationally expensive due to the need to simulate atmospheric transport under

various perturbations. To help with this, we provide these outputs as a separate download at https://doi.org/10.5281/zenodo.4887044 (Bertolacci et al., 2021). These allow the inversions to be done, and for results to be generated, without the need to run the atmospheric transport model. Please see README.md in the code repository for more details.





**Appendix A: Markov chain Monte Carlo algorithm**

As mentioned in Section 2.5, WOMBAT makes inference on the fluxes and the other parameters in the model using a Gibbs

sampler, wherein samples of subsets of the parameters are iteratively drawn from their full conditional distributions (e.g., Tierney, 1994). Recall that the target distribution is $p(\boldsymbol{\alpha}, \boldsymbol{\beta}, \boldsymbol{\kappa}, \boldsymbol{\tau}_w, \boldsymbol{\theta}_{\xi,\epsilon} \mid \mathbf{Z}_2)$, as shown in Eq. (10).

The Gibbs sampler in WOMBAT is as follows. Given the $i$th sample, $\{\boldsymbol{\alpha}^{[i]}, \boldsymbol{\beta}^{[i]}, \boldsymbol{\kappa}^{[i]}, \boldsymbol{\tau}_w^{[i]}, \boldsymbol{\theta}_{\xi,\epsilon}^{[i]}\}$, the $(i+1)$th sample is generated sequentially in the following manner.

1. Sample $\boldsymbol{\alpha}^{[i+1]}$ and $\boldsymbol{\beta}^{[i+1]}$ jointly from $p(\boldsymbol{\alpha}, \boldsymbol{\beta} \mid \boldsymbol{\kappa}^{[i]}, \boldsymbol{\tau}_w^{[i]}, \boldsymbol{\theta}_{\xi,\epsilon}^{[i]}, \mathbf{Z}_2)$.

2. Sample $\boldsymbol{\kappa}^{[i+1]}$ from $p(\boldsymbol{\kappa} \mid \boldsymbol{\tau}_w^{[i]}, \boldsymbol{\alpha}^{[i+1]})$.

3. Sample $\boldsymbol{\tau}_w^{[i+1]}$ from $p(\boldsymbol{\tau}_w \mid \boldsymbol{\kappa}^{[i+1]}, \boldsymbol{\alpha}^{[i+1]})$.

4. Sample $\boldsymbol{\theta}_{\xi,\epsilon}^{[i+1]}$ from $p(\boldsymbol{\theta}_{\xi,\epsilon} \mid \boldsymbol{\alpha}^{[i+1]}, \boldsymbol{\beta}^{[i+1]}, \mathbf{Z}_2)$.

Below, in Appendices A1–A4, we give the details for Steps 1–4. In deriving the conditional distributions, we often make use of the Sherman–Morrison–Woodbury matrix identity and a matrix-determinant lemma (e.g., Henderson and Searle, 1981). The

former identity states that, for conformable matrices $\mathbf{A}$, $\mathbf{U}$, $\mathbf{C}$, and $\mathbf{V}$,

$$(\mathbf{A} + \mathbf{U}\mathbf{C}\mathbf{V})^{-1} = \mathbf{A}^{-1} - \mathbf{A}^{-1}\mathbf{U}(\mathbf{C}^{-1} + \mathbf{V}\mathbf{A}^{-1}\mathbf{U})^{-1}\mathbf{V}\mathbf{A}^{-1},$$

whenever the required inverses exist, while the latter lemma states that

$$|\mathbf{A} + \mathbf{U}\mathbf{C}\mathbf{V}| = |\mathbf{C}^{-1} + \mathbf{V}\mathbf{A}^{-1}\mathbf{U}||\mathbf{C}||\mathbf{A}|.$$

**A1  Sampling $\boldsymbol{\alpha}$ and $\boldsymbol{\beta}$**

Let $\mathbf{B} = (\hat{\boldsymbol{\Psi}}, \mathbf{A})$, $\mathbf{x} = (\boldsymbol{\alpha}', \boldsymbol{\beta}')'$, and $\boldsymbol{\Sigma}_x = \mathrm{bdiag}(\boldsymbol{\Sigma}_\alpha, \sigma_\beta^2 \mathbf{I})$. Then

$$p(\boldsymbol{\alpha}, \boldsymbol{\beta} \mid \boldsymbol{\kappa}, \boldsymbol{\tau}_w, \boldsymbol{\theta}_{\xi,\epsilon}, \mathbf{Z}_2)$$
$$\propto \exp\left[-\frac{1}{2}(\mathbf{Z}_2 - \hat{\mathbf{Z}}_2^0 - \mathbf{B}\mathbf{x})'(\boldsymbol{\Sigma}_\xi + \boldsymbol{\Sigma}_\epsilon)^{-1}(\mathbf{Z}_2 - \hat{\mathbf{Z}}_2^0 - \mathbf{B}\mathbf{x}) - \frac{1}{2}\mathbf{x}'\boldsymbol{\Sigma}_x^{-1}\mathbf{x}\right].$$

The log of this quantity is quadratic in $\mathbf{x}$, and therefore the full conditional distribution of $\mathbf{x}$ is a multivariate Gaussian distribution; specifically,

$$\mathbf{x} \mid \boldsymbol{\kappa}, \boldsymbol{\tau}_w, \boldsymbol{\theta}_{\xi,\epsilon}, \mathbf{Z}_2 \sim \mathrm{Gau}(\boldsymbol{\mu}_x^c, \boldsymbol{\Sigma}_x^c), \tag{A1}$$

where

$$(\boldsymbol{\Sigma}_x^c)^{-1} = \mathbf{B}'(\boldsymbol{\Sigma}_\xi + \boldsymbol{\Sigma}_\epsilon)^{-1}\mathbf{B} + \boldsymbol{\Sigma}_x^{-1},$$





and

$$\boldsymbol{\mu}_x^c = \boldsymbol{\Sigma}_x^c \mathbf{B}'(\boldsymbol{\Sigma}_\xi + \boldsymbol{\Sigma}_\epsilon)^{-1}(\mathbf{Z}_2 - \hat{\mathbf{Z}}_2^0).$$

As described in Section 3, $\boldsymbol{\Sigma}_\epsilon$ is diagonal and, under Markov assumptions, $\boldsymbol{\Sigma}_\xi$ has a sparse inverse. These properties allow us to efficiently compute the required mean and covariance matrix through use of the Sherman–Morrison–Woodbury matrix identity. Once these are computed, sampling from Eq. (A1) is straightforward.

## A2  Sampling $\boldsymbol{\kappa}$

The full conditional distribution of $\boldsymbol{\kappa}$ is

$\quad p(\boldsymbol{\kappa} \mid \boldsymbol{\tau}_w, \boldsymbol{\alpha}) \propto |\boldsymbol{\Sigma}_\alpha|^{-1/2} \exp\left(-\frac{1}{2}\boldsymbol{\alpha}'\boldsymbol{\Sigma}_\alpha^{-1}\boldsymbol{\alpha}\right) p(\boldsymbol{\kappa}),$ (A2)

where, recalling Section 2.4, $p(\boldsymbol{\kappa})$ is a product of beta density functions. Since each $\kappa_j$ and $\tau_{w,j}$ is associated with a spatial region, we also partition $\boldsymbol{\alpha}$ by spatial region. That is, we define $\boldsymbol{\alpha} \equiv ((\boldsymbol{\alpha}^1)', \ldots, (\boldsymbol{\alpha}^{r_s})')'$, where $\boldsymbol{\alpha}^j \in \mathbb{R}^{r_t}$, $j = 1, \ldots, r_s$. For $j = 1, \ldots, r_s$, the $r_t$-dimensional vector $\boldsymbol{\alpha}^j$ can therefore be associated with $\kappa_j, \tau_{w,j}$, and a $r_t \times r_t$ sub-block of the matrix $\boldsymbol{\Sigma}_\alpha^{-1}$, which we denote as $\boldsymbol{\Sigma}_{\alpha,j}^{-1}$. Under the autoregressive model in Section 2.4, $\boldsymbol{\Sigma}_{\alpha,j}^{-1} = \tau_{w,j}\mathbf{Q}_{\alpha,j}$, where

$\quad \mathbf{Q}_{\alpha,j} = \begin{pmatrix} 1 & -\kappa_j & & & \\ -\kappa_j & 1+\kappa_j^2 & \ddots & & \\ & \ddots & \ddots & \ddots & \\ & & \ddots & 1+\kappa_j^2 & -\kappa_j \\ & & & -\kappa_j & 1 \end{pmatrix}, \quad j = 1, \ldots, r_s.$

Since the flux scaling factors in each spatial region are treated as independent *a priori*, Eq. (A2) may be written as $p(\boldsymbol{\kappa} \mid \boldsymbol{\tau}_w, \boldsymbol{\alpha}) = \prod_{j=1}^{r_s} p(\kappa_j \mid \tau_{w,j}, \boldsymbol{\alpha}^j)$, where

$$p(\kappa_j \mid \tau_{w,j}, \boldsymbol{\alpha}^j) \propto |\mathbf{Q}_{\alpha,j}|^{1/2} \exp\left(-\frac{\tau_{w,j}}{2}(\boldsymbol{\alpha}^j)'\mathbf{Q}_{\alpha,j}\boldsymbol{\alpha}^j\right) \kappa_j^{a_{\kappa,j}-1}(1-\kappa_j)^{b_{\kappa,j}-1}.$$ (A3)

To generate samples from Eq. (A2), we therefore successively sample $\kappa_j$, $j = 1, \ldots, r_s$, from its full conditional distribution
(Eq. A3). Equation (A3) does not correspond to any standard distribution, so we use slice sampling (Neal, 2003) to sample from it.

## A3  Sampling $\boldsymbol{\tau}_w$

Similar to $\boldsymbol{\kappa}$, the conditional distribution of $\boldsymbol{\tau}_w$ factorises across spatial regions, and it is therefore given by $p(\boldsymbol{\tau}_w \mid \boldsymbol{\kappa}, \boldsymbol{\alpha}) = \prod_{j=1}^{r_s} p(\tau_{w,j} \mid \kappa_j, \boldsymbol{\alpha}^j)$, where

$\quad p(\tau_{w,j} \mid \kappa_j, \boldsymbol{\alpha}^j) \propto \tau_{w,j}^{r_t/2} \exp\left(-\frac{\tau_{w,j}}{2}(\boldsymbol{\alpha}^j)'\mathbf{Q}_{\alpha,j}\boldsymbol{\alpha}^j\right) \tau_{w,j}^{\nu_{w,j}-1} e^{-\omega_{w,j}\tau_{w,j}}.$ (A4)




The density function in Eq. (A4) is a Gamma density function,

$$\tau_{w,j} \mid \kappa_j, \boldsymbol{\alpha}^j \sim \mathrm{Ga}(\nu_{w,j}^c, \omega_{w,j}^c), \tag{A5}$$

where $\nu_{w,j}^c = \nu_{w,j} + \frac{1}{2} r_t$ and $\omega_{w,j}^c = \omega_{w,j} + \frac{1}{2}(\boldsymbol{\alpha}^j)' \mathbf{Q}_{\alpha,j} \boldsymbol{\alpha}^j$. Therefore, we sample from the full conditional distribution of $\boldsymbol{\tau}_w$ by successively sampling $\tau_{w,j}$, for $j = 1, \ldots, r_s$, directly from Eq. (A5).

## A4  Sampling $\boldsymbol{\theta}_{\xi,\epsilon}$

Since we assume that the parameters governing the correlated and uncorrelated error terms are data-group specific, the full conditional distribution of $\boldsymbol{\theta}_{\xi,\epsilon}$ is

$$p(\boldsymbol{\theta}_{\xi,\epsilon} \mid \boldsymbol{\alpha}, \boldsymbol{\beta}, \mathbf{Z}_2) \propto \prod_{g=1}^{n_g} p(\boldsymbol{\theta}_{\xi_g}, \gamma_g \mid \boldsymbol{\alpha}, \boldsymbol{\beta}, \mathbf{Z}_{2,g}), \tag{A6}$$

where $p(\boldsymbol{\theta}_{\xi_g}, \gamma_g \mid \boldsymbol{\alpha}, \boldsymbol{\beta}, \mathbf{Z}_{2,g}) \propto p(\mathbf{Z}_{2,g} \mid \boldsymbol{\alpha}, \boldsymbol{\beta}, \boldsymbol{\theta}_{\xi_g}, \gamma_g) p(\boldsymbol{\theta}_{\xi_g}, \gamma_g)$; $p(\boldsymbol{\theta}_{\xi_g}, \gamma_g)$ is the joint prior over $\boldsymbol{\theta}_{\xi_g}$ and $\gamma_g$; and

$$p(\mathbf{Z}_{2,g} \mid \boldsymbol{\alpha}, \boldsymbol{\beta}, \boldsymbol{\theta}_{\xi_g}, \gamma_g) \propto |\boldsymbol{\Sigma}_{\xi_g} + \boldsymbol{\Sigma}_{\epsilon_g}|^{-1/2} \exp \Big[ -\frac{1}{2} (\mathbf{Z}_{2,g} - \hat{\mathbf{Z}}_{2,g}^0 - \hat{\boldsymbol{\Psi}}_g \boldsymbol{\alpha} - \mathbf{A}_g \boldsymbol{\beta}_g)' (\boldsymbol{\Sigma}_{\xi_g} + \boldsymbol{\Sigma}_{\epsilon_g})^{-1}$$
$$(\mathbf{Z}_{2,g} - \hat{\mathbf{Z}}_{2,g}^0 - \hat{\boldsymbol{\Psi}}_g \boldsymbol{\alpha} - \mathbf{A}_g \boldsymbol{\beta}_g) \Big]. \tag{A7}$$

The matrix operations in Eq. (A7) may be computed efficiently using the matrix identities given at the start of this section. Since Eq. (A7) does not correspond to any standard distribution, we use slice sampling to sample from it, for $g = 1, \ldots, n_g$.

## Appendix B:  Observation operator for OCO-2 retrievals

The retrieval algorithm used for OCO-2 takes spectral data as input and returns $CO_2$ mole fractions on 20 vertical levels as output via an optimisation procedure. The $CO_2$ mole fractions at these 20 vertical levels are subsequently used to compute a column-averaged $CO_2$ that we associate with a single retrieval. For the $i$th retrieval, denote the vector of retrieved mole fractions as $\mathbf{Z}_{2,i}^r$. Following the argument given by Rodgers and Connor (2003) and Connor et al. (2008), the retrieved mole fractions are given by

$$\mathbf{Z}_{2,i}^r = \mathbf{Y}_{2,i}^{0,r} + \mathbf{A}_i(\mathbf{Y}_{2,i} - \mathbf{Y}_{2,i}^{0,r}) + \boldsymbol{\varepsilon}_i^r,$$

where $\mathbf{Y}_{2,i}^{0,r} = (Y_{2,i,1}^{0,r}, \ldots, Y_{2,i,20}^{0,r})'$ is the vector of prior-mean mole fractions used by the retrieval algorithm, specific to the $i$th retrieval (these are unrelated to the prior mean of the mole-fraction field used in our inversion, $Y_2^0(\cdot, \cdot, \cdot)$); $\mathbf{A}_i$ is the "averaging kernel matrix"; $\mathbf{Y}_{2,i} \equiv (Y_2(\mathbf{s}_i, h_{i,k}, t_i))_{k=1}^{20}$ is the true mole fraction at the 20 vertical levels for the retrieval at geopotential heights $h_{i,k}$, $k = 1, \ldots, 20$; and $\boldsymbol{\varepsilon}_i^r$ is a vector of measurement errors (which may also include systematic biases and errors induced by nonlinearity in the inversion process). The column-averaged retrieval is then

$$Z_{2,i} = \mathbf{c}_i' \mathbf{Z}_{2,i}^r = Y_{2,i}^{0,rc} + \mathbf{c}_i' \mathbf{A}_i(\mathbf{Y}_{2,i} - \mathbf{Y}_{2,i}^{0,r}) + \mathbf{c}_i' \boldsymbol{\varepsilon}_i^r, \tag{B1}$$



where $\mathbf{c}_i \equiv (c_{i,1}, \ldots, c_{i,20})'$ are quadrature weights used to estimate the column average, and $Y_{2,i}^{0,rc} \equiv \mathbf{c}_i' \mathbf{Y}_{2,i}^{0,r}$ is the retrieval prior mean column-averaged $CO_2$. Define $\mathbf{a}_i \equiv (a_{i,1}, \ldots, a_{i,20})'$, where

$$a_{i,k} \equiv \frac{1}{c_{i,k}} (\mathbf{c}_i' \mathbf{A}_i)_k, \quad k = 1, \ldots, 20,$$

and $(\mathbf{c}_i' \mathbf{A}_i)_k$ denotes the $k$th element of $\mathbf{c}_i' \mathbf{A}_i$. The vector $\mathbf{a}_i$ is the "averaging kernel vector" of the $i$th retrieval, which is

available in the official releases of the OCO-2 data. It follows that the observation operator in Eq. (7) is defined as

$$\hat{\mathcal{A}}_i(Y_2(\mathbf{s}_i, \cdot, t_i)) \equiv Y_{2,i}^{0,rc} + \sum_{k=1}^{20} c_{i,k} a_{i,k} \left( Y_2(\mathbf{s}_i, h_{i,k}, t_i) - Y_{2,i,k}^{0,r} \right). \tag{B2}$$

Note that we do not explicitly account for the error term $\mathbf{c}_i' \boldsymbol{\varepsilon}_i^r$ in our definition for $\hat{\mathcal{A}}_i$. This is because it will be absorbed by the error terms we use in the data model (Eq. 8).

The averaging-kernel-vector elements $\{a_{i,k}\}$ are typically close in value to 1. They reflect the fact that the retrieval is not

equally sensitive to the mole fractions at all the vertical levels. At levels where there is less sensitivity (i.e., values $< 1$), the retrieval prior-mean mole fraction will be assigned greater weight when producing the column-average $CO_2$ estimate (Rodgers, 2000).

## Appendix C: Additional figures and tables

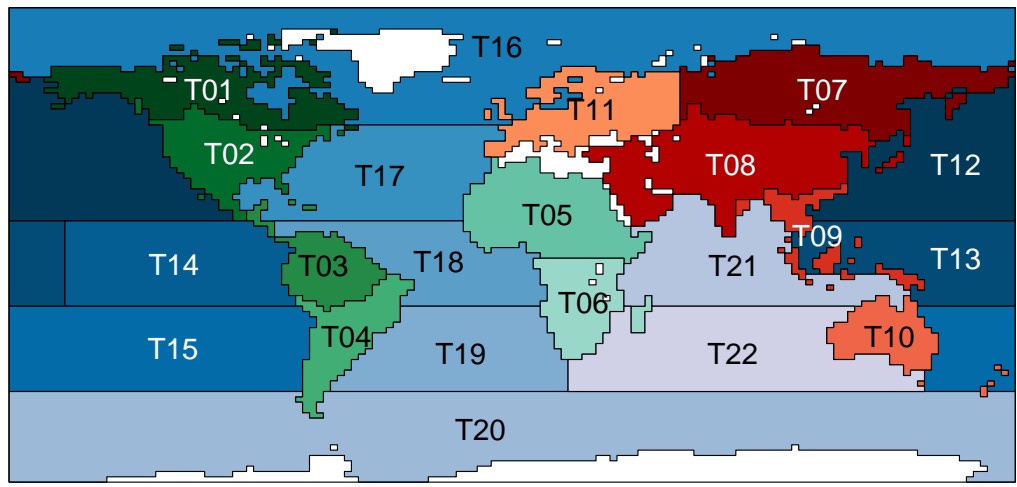

**Figure C1.** Map of the 22 TransCom3 regions used in our study. The names of these regions are given in Table C1. The white regions correspond to areas assumed to have zero $CO_2$ surface flux.





| Code | Name | Type |
|------|------|------|
| T01 | North American Boreal | Land |
| T02 | North American Temperate | Land |
| T03 | Tropical South America | Land |
| T04 | South American Temperate | Land |
| T05 | Northern Africa | Land |
| T06 | Southern Africa | Land |
| T07 | Eurasia Boreal | Land |
| T08 | Eurasia Temperate | Land |
| T09 | Tropical Asia | Land |
| T10 | Australia | Land |
| T11 | Europe | Land |
| T12 | North Pacific Temperate | Ocean |
| T13 | West Pacific Tropical | Ocean |
| T14 | East Pacific Tropical | Ocean |
| T15 | South Pacific Temperate | Ocean |
| T16 | Northern Ocean | Ocean |
| T17 | North Atlantic Temperate | Ocean |
| T18 | Atlantic Tropical | Ocean |
| T19 | South Atlantic Temperate | Ocean |
| T20 | Southern Ocean | Ocean |
| T21 | Indian Tropical | Ocean |
| T22 | South Indian Temperate | Ocean |

**Table C1.** The code, name, and type, of the 22 TransCom3 regions used in our study. A map showing these regions is given in Figure C1.

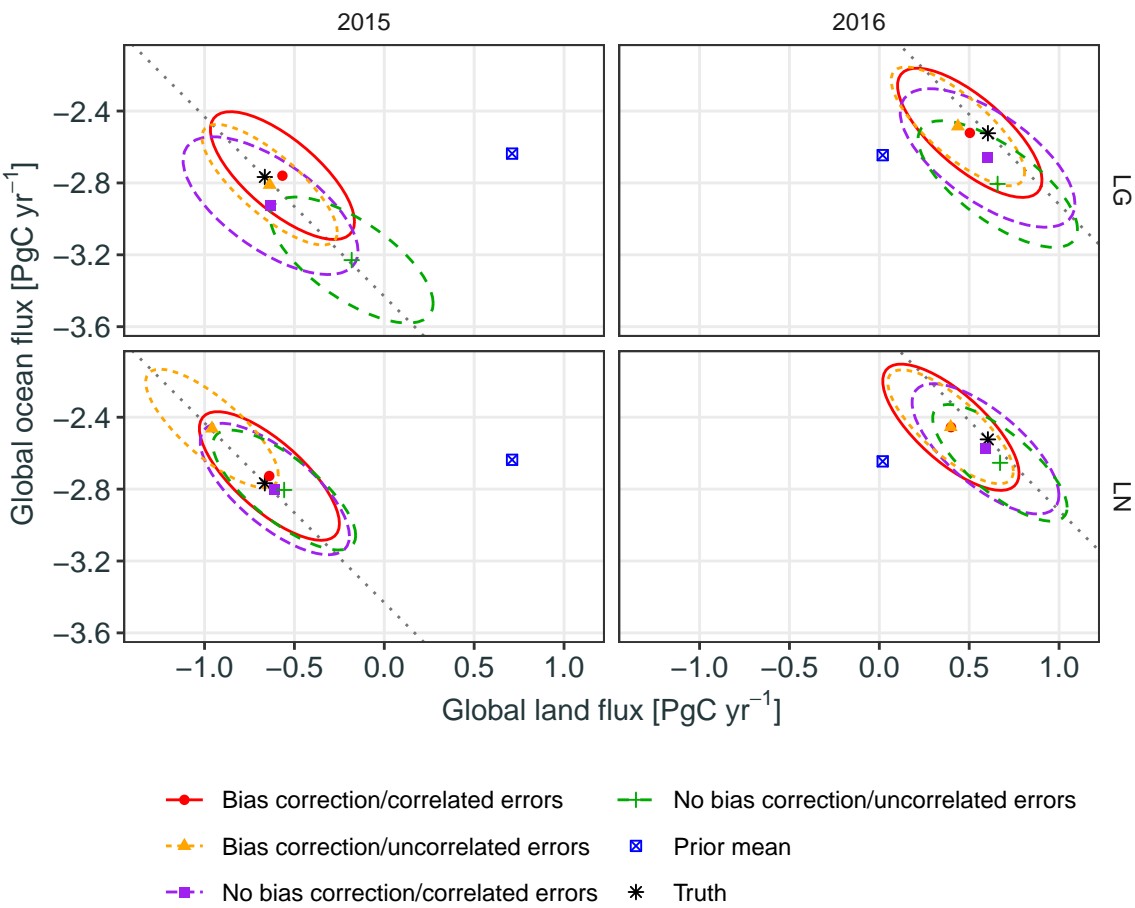

**Figure C2.** As in Figure 3, but for Global ocean versus Global land.







**Figure C3.** As in Figure 3, but for the Southern Africa region (TransCom3 region 06) versus the South American Temperate region (TransCom3 region 04).



**Figure C4.** As in Figure 4, but for zonal bands covering the northern extratropics (23.5°N–90°N), the northern tropics (0°–23.5° N), the southern tropics (23.5° S–0°), and the southern extratropics (90° S–23.5° S).



**Figure C5.** Monthly-averaged TCCON retrievals together with the prior and posterior mole-fraction distributions from WOMBAT, by month and location. Each panel shows the quantities corresponding to a single site. Thick solid black lines show the TCCON retrievals, while thick coloured lines show the prior and posterior mole-fraction means. Shaded areas and thin lines show 95% posterior intervals for the cases "WOMBAT LG" and "WOMBAT LN."





*Author contributions.* AZ-M, MB, and NC designed and wrote the study. JF, AS, MR, and YC contributed to the design and implementation of the GEOS-Chem model simulations. MB and AZM designed and implemented the MCMC algorithm and conducted the analyses.

*Competing interests.* The authors declare that they have no conflict of interest.

*Acknowledgements.* This project was largely supported by the Australian Research Council (ARC) Discovery Project (DP) DP190100180. Andrew Zammit-Mangion's research was also supported by an ARC Discovery Early Career Research Award (DECRA) DE180100203. Noel Cressie's research was also supported by ARC DP150104576. Cressie's and Zammit-Mangion's research was also supported by NASA ROSES grant 17-OCO2-17-0012. Jenny Fisher was supported by ARC DP160101598. Ann Stavert and Matthew Rigby were supported by the UK Natural Environment Research Council grant NE/K002236/1. The TCCON data were obtained from the TCCON Data Archive hosted by CaltechDATA at https://tccondata.org. The authors would also like to thank David Baker for generating the OCO-2 10 seconds average retrievals, and Beata Bukosa, Nicholas Deutscher, Anita Ganesan, Peter Rayner, Andrew Schuh, and members of the OCO-2 Flux Team for invaluable feedback on this research. This work was supported by resources provided by the Pawsey Supercomputing Centre with funding from the Australian Government and the Government of Western Australia. The work was also supported by the NCI National Facility systems at the Australian National University through the National Computational Merit Allocation Scheme supported by the Australian Government.



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
