# Peer review of "WOMBAT v1.0: A fully Bayesian global flux-inversion framework"

_Geoscientific Model Development, 2021_

## Author Comment (AC1)

**Reviewer 1**

**Summary**

*I think the authors have developed a very thoughtful inverse modeling system based on hierarchical Bayesian statistics. The paper is well-written, the statistics are carefully explained, and the model is thoroughly evaluated in the manuscript. I think the inclusion of observational biases, correlated errors, and the calculation of posterior uncertainties are major strengths of the proposed modeling framework. I highly recommend this article for publication.*

**Reply:** We thank Prof. Miller for the constructive comments. We provide replies on a point-by-point basis below.

**Main Comments**

*My one concern is the use of relatively coarse basis functions within the inverse model. On one hand, I understand the necessity of reducing the number of unknowns in the inverse problem. Without the dimension reduction, I doubt it would be computationally possible to estimate the fluxes using an MCMC algorithm (not to mention the number of CTM runs that would be required). On the other hand, I suspect that the quality of the flux estimate will depend on the number and quality of basis functions used in the inverse model. The basis functions used in this study are for relatively large TransCom regions and vary by month, so the inverse modeling framework proposed here would not be able to resolve patterns in fluxes at smaller spatiotemporal scales. This may not be an enormous concern for simulations using OCO-2 but could be a real challenge if using observations that are more sensitive to finer-scale variability in $CO_2$ fluxes (e.g., aircraft observations or observations from future satellites).*

**Reply:** We agree that the use of only a few basis functions spanning large spatial and temporal domains is a limitation of the current setup. The main innovation in this current version of WOMBAT is indeed not the ability to target fine-scale fluxes, but to provide a framework in which uncertainties are identified and integrated at various stages of the inversion model. In the Conclusion (Sect. 6) we acknowledge this and state the following:

> WOMBAT extends a traditional state-space approach to flux inversion, which competes with adjoint-based approaches that allow for a much higher flux dimensionality. [...] However, it would be desirable to have global inversions yield valid inferences at fine spatio-temporal scales globally. Moving forward, WOMBAT will therefore seek to introduce higher dimensionality by using flux basis functions that are at a finer scale than the TransCom3-by-month spatio-temporal basis functions that we have used here, or a fine-scale variation term in the flux process model that can be used to absorb variation in the flux that cannot be explained by the basis functions.

We note, however, that although we could report fluxes at fine resolutions, we only report them at the TransCom3-monthly scale, which is the same resolution as that of our flux scaling factors. We envision WOMBAT to not be problematic in this scenario. We now state in Sect. 6:

> At this chosen resolution we expect WOMBAT to perform well and give predictions that are valid within uncertainty: When one has broad spatial and temporal data coverage of the response functions, as in the case of OCO-2 and a TransCom3-by-month flux resolution, Bayesian synthesis can be expected to be reasonably robust to dimension-reduction error. Further, the WOMBAT posterior distributions over the fluxes are non-Gaussian, and can accommodate skewness and long tails; this added

flexibility mitigates the risk of under-fitting. Moreover, one may introduce additional scaling factors and corresponding basis functions in small "regions of interest," where the fine-scale fluxes are an inferential target, and this is something we are doing in a follow-up iteration of WOMBAT.

*In addition, existing bottom-up models of biospheric $CO_2$ fluxes yield extremely different spatial patterns (e.g., see the MsTMIP model inter-comparison project), and I suspect that the flux estimate might look different if the basis functions were drawn from a biospheric flux model with very different spatiotemporal patterns from CASA.*

**Reply:** Use of different basis functions will certainly yield different flux inferences at the sub-TransCom3 sub-monthly scale, although we believe that inferences over the TransCom3-monthly aggregates would not be markedly different when using data with broad spatial and temporal coverage, partially because of the added model flexibility mentioned in our above response. Further, we recall that in the paper our predictions of out-of-sample TCCON data, which are spatially localised, are often better than those from some models that operate at a much higher native flux resolution. This is an indicator to us that the basis functions used might be of less importance than the incorporation of uncertainties over other parameters in the model and the modelling of error correlation. This is not to say that we do not see the limitations of a low-dimensional flux model, and as indicated above this is an avenue for future work in WOMBAT.

*At a more technical level, the basis functions have different spatial support, and I wonder if that could/should impact the hyperparameters on $\alpha$ (particularly when the hyperparameters are informative).*

**Reply:** While it is true that basis functions have different spatial and temporal resolutions, our scaling factors are at the TransCom3-monthly level, and all our basis functions vary at that, or a finer, resolution. We therefore do not think that the basis-function resolution itself, in this case, should affect our prior distribution on the flux scaling factors. This might need to be a consideration, however, if the flux basis functions are selected such that they have a higher resolution than the inventories from which they are constructed.

*Also, the spatiotemporal patterns of the fluxes within each basis function are assumed known, and I wonder if that assumption could reduce the size of the posterior uncertainty bounds in a way that is undesirable.*

**Reply:** The answer to this is 'yes' at the sub-TransCom3 sub-monthy level if the true flux cannot be reproduced exactly from the weighted basis functions (which is almost certainly the case). The reason for this is that, unlike in typical spatial-statistical low-rank methods, we do not incorporate a fine-scale term in the flux model. The fine-scale term would absorb variation in the flux process that cannot be explained by the basis functions, and to give valid prediction intervals (but not necessarily better predictions) at fine scales; see Zammit-Mangion and Cressie (2021) for an example showing the utility of a fine-scale term in a low-rank model. This term was not included in the current version of WOMBAT for computational reasons. Note that we are not particularly concerned about uncertainty of the flux at the TransCom3-monthly level though, since any unmodelled fine-scale spatio-temporal variation would be averaged out at this level. To address this, we now state in Sect. 6 (already quoted above):

> However, it would be desirable to have global inversions yield valid inferences at fine spatio-temporal scales globally. Moving forward, WOMBAT will therefore seek to introduce higher dimensionality by using flux basis functions that are at a finer scale than the TransCom3-by-month spatio-temporal basis functions that we have used here, or a fine-scale variation term in the flux process model that can be used to absorb variation in the flux that cannot be explained by the basis functions.

*Overall, I think the inverse model is very thoughtfully-constructed, and I don't want to suggest any major changes. With that said, I think the authors may want to head off or preempt some of these concerns within the manuscript. While it is true that some inverse modelers use a 'scaling factor' approach, other inverse modelers do not for the reasons outlined above. Ways to head off these concerns could include (1) provide additional guidance on how to choose basis functions, particularly in light of uncertainties in existing biospheric $CO_2$ flux estimates; (2) discuss the computational feasibility of using a greater number of basis functions (e.g., with an eye toward future studies), and/or (3) include more discussion on the trade-offs between using a greater number of basis functions versus the reduced computational demands of using fewer basis functions. Overall, I think anything that can be done to head off these concerns within the manuscript would strengthen the manuscript.*

**Reply:** Thank you for the suggestion. With regard to Comment (1), we first recall the warning we give to the readers in Sect. 4.2:

> The disadvantage is that the reliance on a priori structures increases the risk of dimension-reduction error because, while our basis functions allow the posterior fluxes to vary at sub-TransCom3-region scales, variations that don't follow the prescribed pattern are necessarily ignored.

We have now also added a suggestion that if inferences on fluxes at specific regions/time spans that are smaller than those resolved by the scaling factors are needed, that additional basis functions are used for those regions/time spans. If this is not possible, then separate inversions with different basis functions might be warranted. We now follow the sentence above with:

> Therefore, if one wishes to make inference at scales that are finer than those resolved by the scaling factors, one should introduce additional basis functions for those regions and time spans. Moreover, for processes where there is disagreement (such as biogeochemical processes), one may consider running separate inversions with basis functions constructed from different inventories and carry out a sensitivity analysis.

Finally, we also acknowledge that there is a large body of work focusing on basis function selection in the context of inversions, and state, following the above sentences:

> We note that there is a considerable body of work tackling basis-function choice in the context of inversion; see, for example, Turner and Jacob (2015).

With regard to Comment (2), we now give a ballpark figure for the maximum number of basis functions that can be used in our model. Specifically, in Sect. 4.6, we now state

> The bottleneck leading to a drastic increase in computing time when modelling correlated errors is due to the term $\hat{\boldsymbol{\Psi}}_g'(\boldsymbol{\Sigma}_{\xi_g} + \boldsymbol{\Sigma}_{\epsilon_g})^{-1}\hat{\boldsymbol{\Psi}}_g$ in Eq. (A7), which needs to be re-evaluated at each MCMC iteration. This operation scales as $O(r^3 + nr^2)$; on hardware current of the year 2021, $r$ needs to be less than 10,000 for computations to remain tractable. On the other hand, when the errors are assumed to be uncorrelated or the length scale parameters are assumed known, many matrix computations can be done once (and not at each MCMC iteration); in this case the bottleneck becomes memory, and on current state-of-the-art servers one may accommodate an order of magnitude more basis functions.

With regard to Comment (3), one should use the largest $r$ possible under the chosen WOM-BAT configuration and the hardware available. Once $r$ is set to its maximum feasible value, a trade-off needs to be made between the spatial resolution and temporal resolution: more subdivisions of space result in fewer subdivisions of time, and vice versa. The strategy employed

will ultimately depend on the purpose of the study; in our case, we chose a monthly temporal resolution and a TransCom3 spatial resolution, as this was the level at which flux esimates were reported in the MIP. We have now included the following suggestion in the conclusion (Sect. 6):

> In the current version of WOMBAT, one should use the largest number of basis functions possible, given available hardware requirements, and find a compromise between the temporal and spatial resolution of the flux basis functions, such that the chosen resolution is as close as possible, or finer, than that at which the flux estimates need to be produced (in our case, this was the TransCom3-by-month level).

**Specific suggestions**

- *Introduction: For many statements in the introduction, there are numerous different citations that you could cite to support the claim. In these cases, you might want to preface the parenthetical citation with "e.g.,". That way, you don't imply that the cited reference is the only reference available that supports this claim. Rather, it is merely one reference, among other possible references.*

  **Reply:** We agree that many citations in the first couple of paragraphs were not suitably prefaced; this has now been fixed.

- *Line 53: Most Bayesian-synthesis inversions use prior distributions to encode uncertainty in the fluxes. Can you be more specific about the innovation in WOMBAT? E.g., do you add in a hyperprior, or some other novel feature?*

  **Reply:** Line 53 was not describing uncertainty on the fluxes themselves, but uncertainty on the *prior beliefs on the fluxes* (for example, priors on the variance parameters), which is not common in standard Bayesian-regularisation inversions. In order to avoid ambiguity we have now added the parenthetic comment "(sometimes referred to as hyperprior distributions)".

- *Lines 58-60: There are a handful of existing papers that have used Monte Carlo simulations or conditional realizations to estimate posterior uncertainties in global trace gas inversions (e.g., papers by Frederic Chevallier and Junjie Liu). However, these papers (to my knowledge) do not use a full-blown MCMC framework nor do they propagate uncertainties in the hyperparameters.*

  **Reply:** Thank you for the comment; Liu et al. (2014) is one such work we are aware of where the uncertainty over the flux is computed from an ensemble of inversions from perturbed prior states and observations. As Prof. Miller points out, this is very different from what we do in WOMBAT, where the uncertainty of the fluxes is affected by that over all the other unknown model parameters. We now state (Sect. 1):

  > We also note that our MCMC framework allows the uncertainties over the fluxes to be affected by uncertainties over all unknown parameters in the model, and it is thus different from the Monte Carlo approach to estimating flux uncertainty used by, for example, Chevallier et al. (2007) and Liu et al. (2014).

- *Line 82: Michalak et al. (2004) did not set the prior flux field to zero everywhere. Rather, the prior flux field was non-informative with an unknown mean that is solved as part of the inverse model.*

  **Reply:** We apologise for this; we now state (beginning of Sect. 2.1):

The field $Y_1^0(\cdot, \cdot)$ could, for example, be treated as a linear regression (e.g., Michalak et al., 2004) or could be constructed using bottom-up estimates of biospheric and/or anthropogenic fluxes (e.g., Basu et al., 2013).

- *Line 101: Should there be a comma after "WOMBAT"?*

  **Reply:** We agree that it reads better with the comma.

- *Sect. 2.2: It feels slightly confusing to use the variable "Y" to denote both fluxes and atmospheric mole fractions.*

  **Reply:** It is standard in multivariate spatial/spatio-temporal modelling to denote the two processes of interest being jointly modelled (in this case the fluxes and the atmospheric mole fractions) using the symbol $Y$, and to use subscripts to distinguish between the two; see, for example, Gneiting et al. (2010, above Eq. 2) and Cressie and Zammit-Mangion (2016, Eq. 1).

- *Line 172: I believe the column average is also based on the pressure weighting function (not just the averaging kernel).*

  **Reply:** The averaging kernel vector, which is what we are implicitly referring to here, incorporates within it the pressure weighting function; see, for example, Connor et al. (2008, Eq. 8). We give full details, including the distinction between the averaging kernel matrix and the averaging kernel vector, in Appendix B. These details are left in the appendix as the retrieval operation is not of central importance to our discussion.

- *Line 202: What do you mean by "statistical efficiency"?*

  **Reply:** By statistical efficiency we mean the ability of an estimator derived from the posterior distribution (e.g., the posterior mean) to have a low mean-squared prediction error with respect to the true (unknown) value. Since this technical description would detract from the main message we now simply state (end of Sect. 2.3)

  > predictions of the flux worsen under the assumption of uncorrelated errors if the errors truly are correlated.

- *Line 225: Michalak et al. also estimated uncertainties in the hyperparameters but did not explicitly integrate those uncertainties in an inverse model. They also did not use a hyperprior. In my experience, some parameters are relatively easy to estimate using maximum-likelihood (ML) (e.g., the variances of the covariance matrices) while others are challenging to estimate from the atmospheric observations using ML (e.g., the covariance structure). I.e., often there isn't sufficient information in the atmospheric observations to guide the choice of these parameters. I think one advantage of using a hyperprior is that there is prior information to inform the hyperparameters even when the atmospheric observations are not sufficient to inform those parameters.*

  **Reply:** We concur with Prof. Miller that, while Michalak et al. (2005) did give expressions for the approximate variance of the maximum likelihood estimators, these uncertainties were not propagated to the fluxes. This is, therefore, different to what is done in WOMBAT, where uncertainties in all unknown parameters are propagated to those over the fluxes. We also agree that Michalak et al. (2005) did not make use of a hyperprior distribution (a

uniform hyperprior is implied). Our statement "The use of prior distributions on $\boldsymbol{\kappa}$ and $\boldsymbol{\tau}_w$ adds an extra level of flexibility and allows the modeller to express the "uncertainty on the uncertainties" in an inversion framework" stresses that one can encode the prior uncertainty over these parameters as needed. We have now added the following sentence to stress the advantage suggested by the reviewer, namely that the prior distributions can be used to supply important information regarding the unknown parameters when the observations are not sufficiently informative (Sect. 2.4):

> A related advantage is that the prior distributions can be used to provide information on the variance parameters even when the mole-fraction observations are not informative of the parameters.

- *Lines 369-370: I don't necessarily agree with the statement that the space-time patterns in existing bottom-up estimates are realistic. Bottom-up biospheric flux estimates from recent inter-comparison projects show widely differing spatial and seasonal patterns. For example, there are large spatial and seasonal differences among models participating in both MsTMIP and the TRENDY model comparisons. The large differences among existing bottom-up biospheric flux models implies that regional space-time $CO_2$ flux patterns are not well known.*

  **Reply:** Thank you and we agree. We chose the word "realistic" as these space-time patterns often are motivated by physical principles, and thus they are a marked improvement over the space-time patterns of generic basis functions that are commonly used in spatial statistics (e.g., Wikle et al., 2019, Figure 4.7) such as wavelets or bisquare functions. We now state (Sect. 4.2):

  > However, the space-time patterns are dictated by those in the inventories used to construct the basis functions. Although there is a general lack of agreement between inventories targeting the same processes (e.g., Huntzinger et al., 2017), these spatio-temporal patterns would not be unreasonable and certainly more reasonable than those from generic basis functions commonly used in spatial statistics (e.g., Wikle et al., 2019, Chapter 4).

- *Line 370: I wouldn't say that this assumption is ubiquitous. While some inverse modeling systems do make this assumption, there are also plenty of inverse modeling systems that do not make this assumption. For example, inverse modeling systems like the CMS-Flux system at NASA JPL and CarbonTracker-Lagrange system at the NOAA Global Monitoring Laboratory do not make this assumption.*

  **Reply:** Thank you, the word 'ubiquitous' was inappropriate in this case. We now say that this assumption is "often made."

- *Line 465: The word "embarrassingly" feels a bit too colloquial here.*

  **Reply:** "Embarrassingly parallel" is not colloquial in this case, but an accepted technical term that is used to describe a process that can be easily split up into tasks that can be executed in parallel.

- *Fig. 3 caption (and other figures throughout the manuscript): I think it would be useful to provide the main takeaway message of the figure within the caption. I think that would help guide the reader on how to interpret the results.*

**Reply:** While we see the potential benefit of having a takeaway message, it is a stylistic choice that we use in all our articles to not provide interpretations in the caption, and only provide details on what the image is showing. The main reason for this is that there is usually a lot of context surrounding the interpretation or the main message of a figure, which is best conveyed in the main text.

- *Line 558: I think there's a typo at the beginning of this line.*

  **Reply:** We are sorry but we could not locate the typo.

- *Line 582: I would use a more specific term than "deeper" to avoid confusion here. (This wording is also used later in the manuscript.)*

  **Reply:** Thank you for the suggestion. We have now replaced "deeper posterior-mean sink" with "larger posterior-mean sink", and "deepest sink in the cycle" with "largest sink in the cycle."

- *Fig. 4: The shaded areas on the right hand side of this plot are difficult to distinguish from one another. I wonder if there is a way to improve this aspect of the visualization. This suggestion applies to a few other similar figures as well.*

  **Reply:** We agree and now show only the global land/ocean LG results in Fig. 4, which we reproduce here in this Response as Fig. R1. We have put the LN results for global land/ocean in the supplement. Similarly, in the supplement, we now also show the LG/LN zonal results in separate figures.

- *Overall: I think it would be nice to have an appendix with all of the variable definitions, so the reader can keep track of what's what without searching through the manuscript text.*

  **Reply:** Thank you for the suggestion. Instead of an appendix, we have now extended Fig. 1 (Fig. R2 in this Response) to include the processes, and also in the accompanying text we include definitions of all the symbols.

**Reviewer 2**

**Comments**

*This paper is very well presented and interesting and contributes significantly to the inversion community. I support publication after the following concerns are addressed:*

1. **Supplement vs appendix:** *The appendix are too abundant. Please keep only Appendix A-B and put the rest in supplement*

   **Reply:** Thank you for the suggestion, we have now put the additional tables and figures in a supplement.

2. **Better identify advantage of WOMBAT:** *Even though the OSSE and MIP comparison are interesting, it is hard fully separate the impact of WOMBAT compared to classical inversions and part of the patterns are necessarily linked to the CTM used. Using another CTM is beyond the scope of the manuscript, but I would suggest showing an analytical inversion with the same base functions, but no estimation of hyperparameters. It would*

[Figure]

Figure R1: Annual (left column) and monthly (right column) fluxes for the globe (first row), land (second row), and ocean (third row). Summaries of flux estimates from the MIP and the flux estimate from WOMBAT are shown, split into the prior and LG inversions. Thick lines represent the ensemble means for the MIP and the (prior or posterior) means for WOMBAT. Shaded areas and thin lines for the MIP represent values between the ensemble minimum and maximum, while for WOMBAT they represent values in the 95% credible intervals (cred. int.). Fossil-fuel fluxes are excluded from all figures. Note that each row of plots has a different vertical scale.

[Figure]

Figure R2: Graphical model summarising the relationship between the variables, processes, and parameters to be inferred, and the grouped data $\{\mathbf{Z}_{2,g} : g = 1, \ldots, n_g\}$.

*show how a classical inversion would perform with GEOS-Chem compared to WOMBAT, which would partly disentangle different error sources*

**Reply:** Thank you for the suggestion. We have now added a case study to Sect. 5.1 which reflects the classical approach to inversion. Specifically, we have added to the four existing setups one that assumes that errors are uncorrelated and unbiased, and all the hyperparameters are fixed to their true values in the OSSE. No MCMC is required for this classical inversion setup. Even though this setup has the benefit of hyperparameters being fixed to their true values, flux estimates are still worse than those from those that do model biases and/or correlated errors. This further shows the importance of modelling and estimating these quantities when they are indeed present. The new version of Table 3, which includes results from this additional setup, is also given in this Response as Table R1.

3. **Information content:** *The author should further elaborate on the information content in the system. In particular, contrary to classical inversions, WOMBAT must estimate hyperparameters, which are much more numerous than in standard inversions. Thus, is there enough observations? When adding new regions is conflicting with the number of observations?*

**Reply:** While the model used in WOMBAT appears to be highly-parameterised, this is not the case. In our inversion we only had $r = 682$ flux scaling factors to estimate, $r_s = 22$ autoregressive parameters; $r_s = 22$ flux innovation precisions; one error variance inflation factor; one error length scale; and one error proportion; and three measurement-bias coefficients, to yield a total of 732 unknown quantities that need to be estimated. On the other hand we had 114,808 LG and 129,203 LN OCO-2 retrievals with which to do the inversion, orders of magnitude more than the number of unknown parameters in our model. While these retrievals are not independent, the information contained within them is substantial and, as we show in our OSSE in Sect. 5.1, enough to yield posterior distributions that are practically useful. In Sect. 4.6 we now state:

> In our studies, we found that the vast majority of posterior distributions were different from the prior distributions, and this is not surprising. Although complex, the model used in WOMBAT is low-dimensional, and in this setup the total number of unknowns is 732 ($r = 682$ of which are flux scaling factors), which is orders of magnitude less than the number of LG and LN data available for the

Table R1: Root-mean-squared error (RMSE) and continuous ranked probability score (CRPS) when estimating monthly regional fluxes using LG and LN data in the OSSE of Sect. 5.1. The lower the error or the score, the better the performance. Five setups in WOMBAT are evaluated, and the regions and time periods over which these summaries (averages) are obtained are the same as those used for constructing flux basis functions (see Sect. 4.2).

| | RMSE [PgC mo$^{-1}$] | | CRPS | |
| Setup | LG | LN | LG | LN |
|---|---|---|---|---|
| Bias correction/correlated errors | 0.023 | 0.021 | 0.010 | 0.009 |
| Bias correction/uncorrelated errors | 0.038 | 0.038 | 0.015 | 0.016 |
| No bias correction/correlated errors | 0.045 | 0.026 | 0.016 | 0.011 |
| No bias correction/uncorrelated errors | 0.092 | 0.063 | 0.034 | 0.026 |
| No bias correction/uncorrelated errors/fixed hyperparams | 0.052 | 0.039 | 0.022 | 0.017 |
| Prior | 0.036 | 0.036 | 0.018 | 0.019 |

inversion (114,808 and 129,203, respectively). Specifically, these data prove to be highly informative of the unknown parameters in our model.

An attractive property of Bayesian inference is that when the data are not informative of a parameter within the model, it yields a posterior distribution that is similar to the prior distribution. If the latter is uninformative, then the posterior distribution will also be uninformative. Therefore, one needs not to be overly concerned as to 'wrong' estimates if one adds a new region that is not adequately observed, say. In the paper we now state (Sect. 4.6):

> Generally, one need not be overly concerned if a parameter is poorly constrained by the data. In such cases, a Bayesian framework such as WOMBAT returns a posterior distribution over the poorly constrained parameter that tends toward the prior distribution, which in turn encapsulates the a priori belief on the plausible range of values the parameter can take.

*What is the cost of one MCMC with the present configuration, and when does it get too expensive to compute with increasing dimension?*

**Reply:** In the last paragraph of Sect. 4.6 we give the computational details: We say that we require 15 minutes for 11,000 iterations if the errors are assumed to be uncorrelated, and two hours for 11,000 iterations if the errors are assumed to be correlated and the error length scales need to be estimated. This equates to 0.08 s and 0.655 s per iteration, respectively. We have now also added the following computational details to this paragraph:

> The bottleneck leading to a drastic increase in computing time when modelling correlated errors is due to the term $\hat{\boldsymbol{\Psi}}_g'(\boldsymbol{\Sigma}_{\xi_g} + \boldsymbol{\Sigma}_{\epsilon_g})^{-1}\hat{\boldsymbol{\Psi}}_g$ in Eq. (A7), which needs to be re-evaluated at each MCMC iteration. This operation scales as $O(r^3 + nr^2)$; on hardware current of the year 2021, $r$ needs to be less than 10,000 for computations to remain tractable. On the other hand, when the errors are assumed to be uncorrelated or the length scale parameters are assumed known, many matrix computations can be done once (and not at each MCMC iteration); in this case the bottleneck becomes memory, and on current state-of-the-art servers one may accommodate an order of magnitude more basis functions.

Please also see our responses to Reviewer 1 (*Main Comments*) for more comments on computations and basis-function choice.

**References**

Basu, S., Guerlet, S., Butz, A., Houweling, S., Hasekamp, O., Aben, I., Krummel, P., Steele, P., Langenfelds, R., Torn, M., Biraud, S., Stephens, B., Andrews, A., and Worthy, D. (2013). Global $CO_2$ fluxes estimated from GOSAT retrievals of total column $CO_2$. *Atmospheric Chemistry and Physics*, 13:8695–8717.

Chevallier, F., Bréon, F.-M., and Rayner, P. J. (2007). Contribution of the Orbiting Carbon Observatory to the estimation of $CO_2$ sources and sinks: Theoretical study in a variational data assimilation framework. *Journal of Geophysical Research: Atmospheres*, 112(D9).

Connor, B. J., Boesch, H., Toon, G., Sen, B., Miller, C., and Crisp, D. (2008). Orbiting Carbon Observatory: Inverse method and prospective error analysis. *Journal of Geophysical Research: Atmospheres*, 113(D5).

Cressie, N. and Zammit-Mangion, A. (2016). Multivariate spatial covariance models: a conditional approach. *Biometrika*, 103:915–935.

Gneiting, T., Kleiber, W., and Schlather, M. (2010). Matérn cross-covariance functions for multivariate random fields. *Journal of the American Statistical Association*, 105:1167–1177.

Huntzinger, D., Michalak, A., Schwalm, C., Ciais, P., King, A., Fang, Y., Schaefer, K., Wei, Y., Cook, R., Fisher, J., Hayes, D., Huang, M., Ito, A., Jain, A., Lei, H., Lu, C., Maignan, F., Mao, J., Parazoo, N., Peng, S., Poulter, B., Ricciuto, D., Shi, X., Tian, H., Wang, W., Zeng, N., and Zhao, F. (2017). Uncertainty in the response of terrestrial carbon sink to environmental drivers undermines carbon-climate feedback predictions. *Scientific Reports*, 7:1–8.

Liu, J., Bowman, K. W., Lee, M., Henze, D. K., Bousserez, N., Brix, H., James Collatz, G., Menemenlis, D., Ott, L., Pawson, S., Jones, D., and Nassar, R. (2014). Carbon monitoring system flux estimation and attribution: impact of ACOS-GOSAT $X_{CO_2}$ sampling on the inference of terrestrial biospheric sources and sinks. *Tellus B: Chemical and Physical Meteorology*, 66:22486.

Michalak, A. M., Bruhwiler, L., and Tans, P. P. (2004). A geostatistical approach to surface flux estimation of atmospheric trace gases. *Journal of Geophysical Research: Atmospheres*, 109(D14).

Michalak, A. M., Hirsch, A., Bruhwiler, L., Gurney, K. R., Peters, W., and Tans, P. P. (2005). Maximum likelihood estimation of covariance parameters for Bayesian atmospheric trace gas surface flux inversions. *Journal of Geophysical Research: Atmospheres*, 110(D24).

Turner, A. and Jacob, D. (2015). Balancing aggregation and smoothing errors in inverse models. *Atmospheric Chemistry and Physics*, 15:7039–7048.

Wikle, C. K., Zammit-Mangion, A., and Cressie, N. (2019). *Spatio-Temporal Statistics with R.* Chapman & Hall/CRC Press, Boca Raton, FL.

Zammit-Mangion, A. and Cressie, N. (2021). FRK: An R package for spatial and spatio-temporal prediction with large datasets. *Journal of Statistical Software*, 98(4):1–48.